# Distinct adaptive mechanisms drive recovery from aneuploidy caused by loss of the Ulp2 SUMO protease

Hong-Yeoul Ryu[1], Francesc López-Giráldez [2], James Knight[2], Soo Seok Hwang [3], Christina Renner[4], Stefan G. Kreft[4] & Mark Hochstrasser[1]

In response to acute loss of the Ulp2 SUMO-specific protease, yeast become disomic for chromosome I (ChrI) and ChrXII. Here we report that ChrI disomy, which creates an adaptive advantage in part by increasing the dosage of the Ccr4 deadenylase, was eliminated by extended passaging. Loss of aneuploidy is often accompanied by mutations in essential SUMO-ligating enzymes, which reduced polySUMO-conjugate accumulation. The mRNA levels for almost all ribosomal proteins increase transiently upon initial loss of Ulp2, but elevated Ccr4 levels limit excess ribosome formation. Notably, extended passaging leads to increased levels of many small nucleolar RNAs (snoRNAs) involved in ribosome biogenesis, and higher dosage of three linked ChrXII snoRNA genes suppressed ChrXII disomy in *ulp2Δ* cells. Our data reveal that aneuploidy allows rapid adaptation to Ulp2 loss, but long-term adaptation restores euploidy. Cellular evolution restores homeostasis through countervailing mutations in SUMO-modification pathways and regulatory shifts in ribosome biogenesis.

[1] Department of Molecular Biophysics & Biochemistry, Yale University, New Haven, CT 06520, USA. [2] Yale Center for Genome Analysis, Yale University, New Haven, CT 06520, USA. [3] Department of Immunobiology, Yale University, New Haven, CT 06520, USA. [4] Department of Biology, Molecular Microbiology, University of Konstanz, Universitaetsstrasse 10, 78457 Konstanz, Germany. Correspondence and requests for materials should be addressed to M.H. (email: mark.hochstrasser@yale.edu)

Environmental changes induce many acute and long-term adaptive cellular responses. Mammals have diverse adaptive mechanisms for stress resistance, such as immune responses to pathogens or hormone-mediated homeostatic processes[1,2]. At the cellular level, acute changes in the local microenvironment, such as changes of pH or temperature, oxidative stress, or nutrient limitation, may induce programmed cell death or trigger adaptive changes that include gene mutation, aneuploidy, changes in gene expression or epigenetic alterations[3–5]. Aneuploidy alters the dosage of genes on the extra chromosome(s), often amplifying a particular gene(s) required to increase cell survival[3]. In some cases, rapidly activated mechanisms that restore homeostasis are superseded by more robust, long-term adaptations[6]. For instance, specific aneuploidies are rapidly induced by heat or high pH; however, persistent exposure to the same stresses eventually leads to gene mutation and alterations in transcription, instead of aneuploidy[7], which is usually deleterious to cell fitness but is a common feature of cancer cells[8].

The yeast Saccharomyces cerevisiae has been a useful model for studying evolutionary processes under laboratory conditions, providing insights into adaptive genetic mechanisms in microbial populations subject to specific growth conditions[9]. Adaptations often involve genomic sequence polymorphisms, including single-nucleotide variations, insertions, deletions, and other structural variations, as well as large copy number variants and aneuploidy[10–12]. Such studies revealed that many genome modifications contribute to species diversification, evolution, and adaptation[4,13,14].

The small ubiquitin-like modifier (SUMO) protein is a conserved post-translational modifier that regulates a host of proteins in eukaryotic cells[15]. The C-terminus of mature SUMO is first activated by the heterodimeric SUMO-activating enzyme E1 (Aos1/Uba2), which then transfers the SUMO to the SUMO-conjugating enzyme E2 (Ubc9). Finally, SUMO is transferred to specific substrates with the aid of an E3 SUMO ligase (Siz1, Siz2, or Mms21). SUMO-substrate deconjugation in yeast is catalyzed by the SUMO proteases Ulp1 and Ulp2[16]. While much is known about Ulp1, most molecular functions of Ulp2 remain unclear. Ulp2 has particularly high activity towards polySUMO chains, and it preferentially localizes to the nucleus. Its loss results in a pleiotropic mutant phenotype including defects in cell growth; sensitivity to heat, DNA damage or aberrant spindle formation; and high rates of chromosome and plasmid loss[17]. Recently, we reported that acute loss of Ulp2 leads to aneuploidy, specifically, a double disomy of chromosome I (ChrI) and ChrXII, an adaptive mechanism that compensates for a severely dysregulated SUMO system[18]. Increased dosage of two ChrI genes, CLN3, encoding a G1 cyclin, and CCR4, encoding the catalytic deadenylase subunit of the Ccr4-Not complex, were shown to drive the ChrI disomy[18,19].

Yeast SUMO pathway mutants show abnormal formation of 60s pre-ribosomes[20]. In normal ribosome biogenesis, the 35s ribosomal RNA (rRNA) precursor is processed into the mature 25s, 18s, and 5.8s rRNAs; the 5s rRNA is independently transcribed[21]. The 60s subunit contains 46 proteins and the 25s, 5.8s, and 5s rRNAs, while the 40s subunit comprises the 18s rRNA and 33 proteins. Ribosome biogenesis involves many rRNA modifications that are catalyzed by small nucleolar ribonucleoprotein complexes (snoRNPs) composed of small nucleolar RNAs (snoRNAs) and nucleolar proteins[22]. The 77 snoRNAs in S. cerevisiae fall into three classes: box C/D snoRNAs responsible for 2′-O-ribose methylation, box H/ACA snoRNAs mediating pseudouridylation, and the NME1 snoRNA that participates in rRNA cleavage. SUMO is also involved in snoRNP biogenesis through its conjugation to Nop58 and Nhp2, core proteins of both Box C/D and H/ACA snoRNPs[23].

Here, we report that the ChrI and ChrXII aneuploidy triggered by acute loss of Ulp2 is completely lost within 500 generations of growth. RNA-seq experiments reveal that many RP gene transcripts are substantially increased in response to the potentially lethal effects of acute ULP2 gene loss. However, following extended culture, RP mRNAs subside to wild-type (WT) levels while more than half of the snoRNAs are significantly upregulated. Provision of high levels of a polycistronic snoRNA cluster, consisting of SNR61, SNR55, and SNR57 from ChrXII, suppresses the enhanced expression of RPs in cells lacking ULP2. Linking the two disomies in nascent ulp2Δ cells, the ChrI duplication drives an increase in Ccr4 binding at snoRNA and rDNA loci, preventing their overexpression. The ulp2Δ cells also acquire mutations in genes encoding the SUMO-ligating enzymes Ubc9 or Uba2/Aos1 during laboratory evolution. The point mutations in these essential genes inhibit the accumulation of high-molecular-weight (HMW) polySUMO conjugates caused by loss of Ulp2. This inhibition of HMW polySUMO is paralleled by suppression of ulp2Δ growth defects. Taken together, our data suggest that ribosome biogenesis is an essential target of SUMO regulation, and its perturbation by loss of the polySUMO-selective Ulp2 protease is compensated by distinct short-term and long-term adaptive mechanisms. Mutations in essential SUMO-conjugation pathway components are a recurrent feature of long-term adaptation as well. This remarkable genetic malleability in response to disruption of SUMO-protein conjugation dynamics is likely relevant to other pathways that link cell growth and division to essential post-translational modifications.

## Results

**ChrI disomy of ulp2Δ cells is lost after extended culture**. We first explored how quickly aneuploidy is established. For this we developed an improved inducible AID* degron-tagged ULP2 construct in which the degradation of the mRNA can also be induced through a neomycin-regulated self-cleaving hammerhead ribozyme in the 3′-UTR (Supplementary Fig. 1). We observed that a single ChrI disomy was preferentially generated in cells after depletion of Ulp2, although it took many days in culture. Although we could no longer detect the Ulp2 protein even at the first timepoint (2 days), it is likely that there was still a very low level present that limited the transition to the aneuploid state. Potentially, the transition occurs in a step-wise fashion, with ChrI disomy needed before ChrXII duplication can occur.

We had reported that ChrXII, but not ChrI, disomy was eliminated after ~100 generations of continuous culture of ulp2Δ cells with repeated cycles of dilution and outgrowth in fresh medium[18]. To determine if the ChrI disomy would eventually also be lost during laboratory evolution, we monitored the relative copy number of ChrI at various points after the establishment of the ulp2Δ null state (Fig. 1a, b). We previously analyzed copy number for up to 250 cell generations and had not seen loss of ChrI[18]. By contrast, the extra copy of ChrI in ulp2Δ cells had disappeared by 500 generations. The more rapid elimination of extra ChrXII compared to ChrI, and the eventual loss of both extra copies, was verified by an independent experiment (Supplementary Fig. 2). Yeast ulp2Δ mutants are impaired in cell-cycle progression and chromosome segregation as well as growth[17,24]. The growth and cell-cycle defects were also fully suppressed in the long-term adapted cells (Fig. 1c, e), which was paralleled by a (partial) reduction in HMW SUMO conjugates (Fig. 1d). Hence, adaptation to loss of Ulp2 leads to a transient aneuploidy, which after culturing for many generations results in reversion to euploidy, nearly WT growth and cell-cycle progression, and reduced accumulation of the HMW SUMO conjugates typical of ulp2Δ cells.

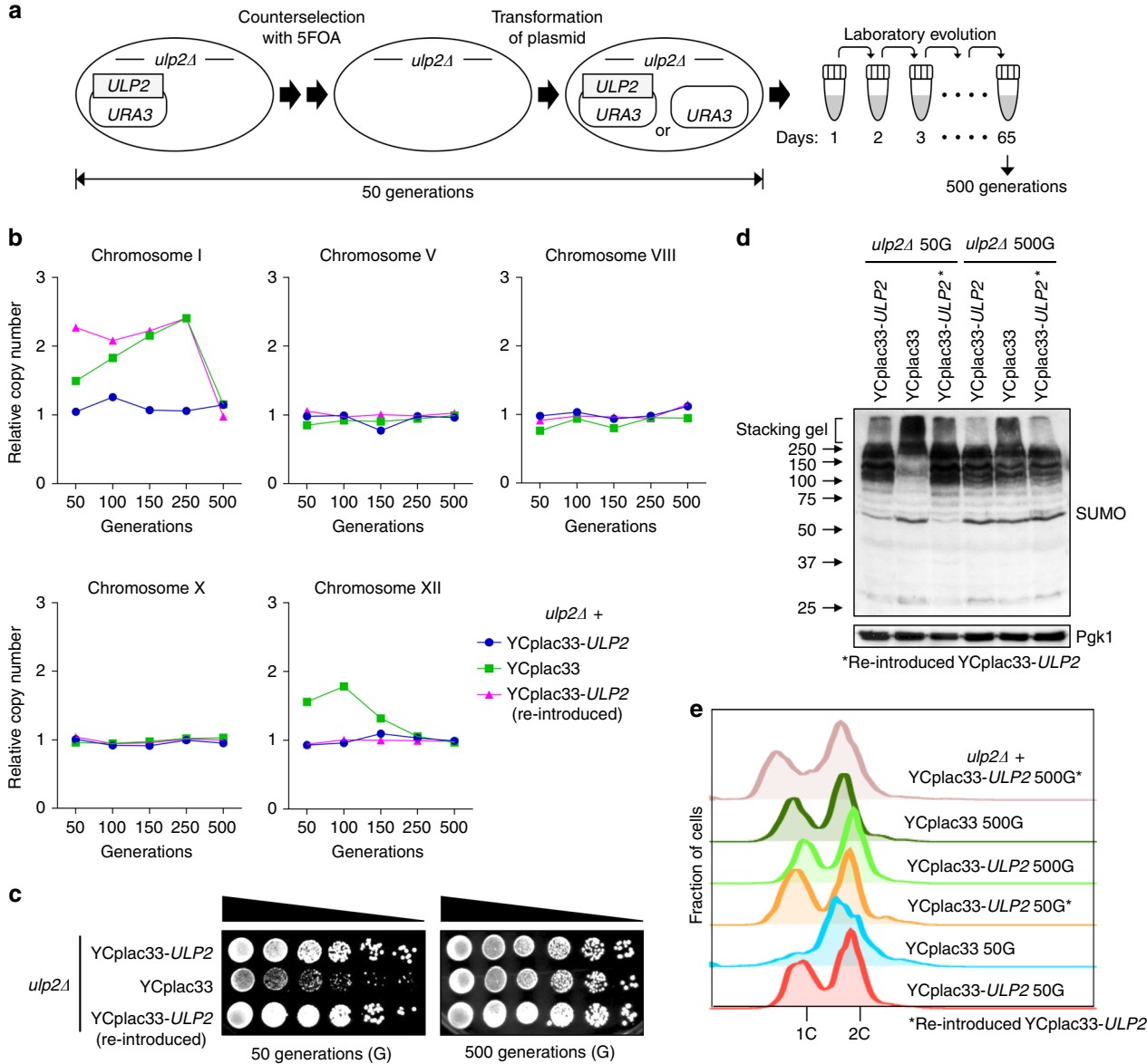

**Fig. 1** Aneuploidy of *ulp2Δ* cells can be reversed by laboratory evolution. **a** Scheme for creation of nascent *ulp2Δ* cells and their subsequent passaging. The *ulp2Δ* cells containing YCplac33-*ULP2* were streaked on SD + FOA plates twice sequentially to evict the YCplac33-*ULP2* plasmid, and they were then transformed either with YCplac33-*ULP2* or YCplac33. Cells grew for ~50 generations (50 G) during these procedures. For long-term passaging, cells were grown until saturation and then diluted 1:120 in fresh YPD (6.9 generations per dilution). This process was repeated daily. **b** Chromosome copy number of *ulp2Δ* cells with the indicated plasmids was monitored by qPCR assay at the indicated number of generations. Copy number was determined relative to WT (MHY500 + YCplac33) cells. Chromosome copy number between 50 and 250 generations were monitored previously[18], and the data are included here. **c** Growth of the indicated *ulp2Δ* strains at 50G and 500G. After spotting cells in fivefold serial dilutions, the YPD plates were incubated for 2 days at 30 °C. **d** Immunoblot analysis of sumoylated proteins in extracts prepared from the strains in **c**. Anti-Pgk1 blotting was used to verify the equal loading. Asterisks indicate reintroduced YCplac33-*ULP2*. The stacking gel (bracket) and molecular size standards are indicated. **e** Flow cytometry of DNA content of the strains from **c**. 1C and 2C indicate unreplicated DNA (haploid) and replicated DNA (diploid), respectively

**High RP levels in *ulp2Δ* normalize after multiple generations**. Aneuploidy is often associated with responses to stress; the duplicated chromosomes encode genes whose increased levels appear to counteract detrimental effects of the stress[3,25]. Gene expression imbalances resulting from aneuploidy, however, carry a fitness cost[26,27]. Aneuploidy may serve as a rapid stopgap response that can be superseded by more sustainable adaptations that somehow counter the stress[7]. Because aneuploidy disappeared and growth and cell-cycle defects were mitigated by evolving *ulp2Δ* cells over long periods (Fig. 1), we asked if the evolved strain had acquired any genetic changes.

Genome-wide gene expression profiles of the evolved *ulp2Δ* cells were analyzed by RNA-seq. In these comparisons, the *ulp2Δ* cells either carried a WT *ULP2* plasmid or empty vector and were passaged for 50 or 500 generations (G). Expression of many transcripts was significantly different among the indicated strains (Supplementary Figs. 3a and 4). Differentially expressed genes and their changes relative to expression in the WT control strain (*ulp2Δ* + YCplac33-*ULP2*, 50 G) are shown in Fig. 2a. In nascent *ulp2* null cells (*ulp2Δ* + YCplac33, 50 G), ~80% of the significant changes (twofold or more) involved increases in transcript levels (Fig. 2a and Supplementary Fig. 3b). Genes encoding ribosomal

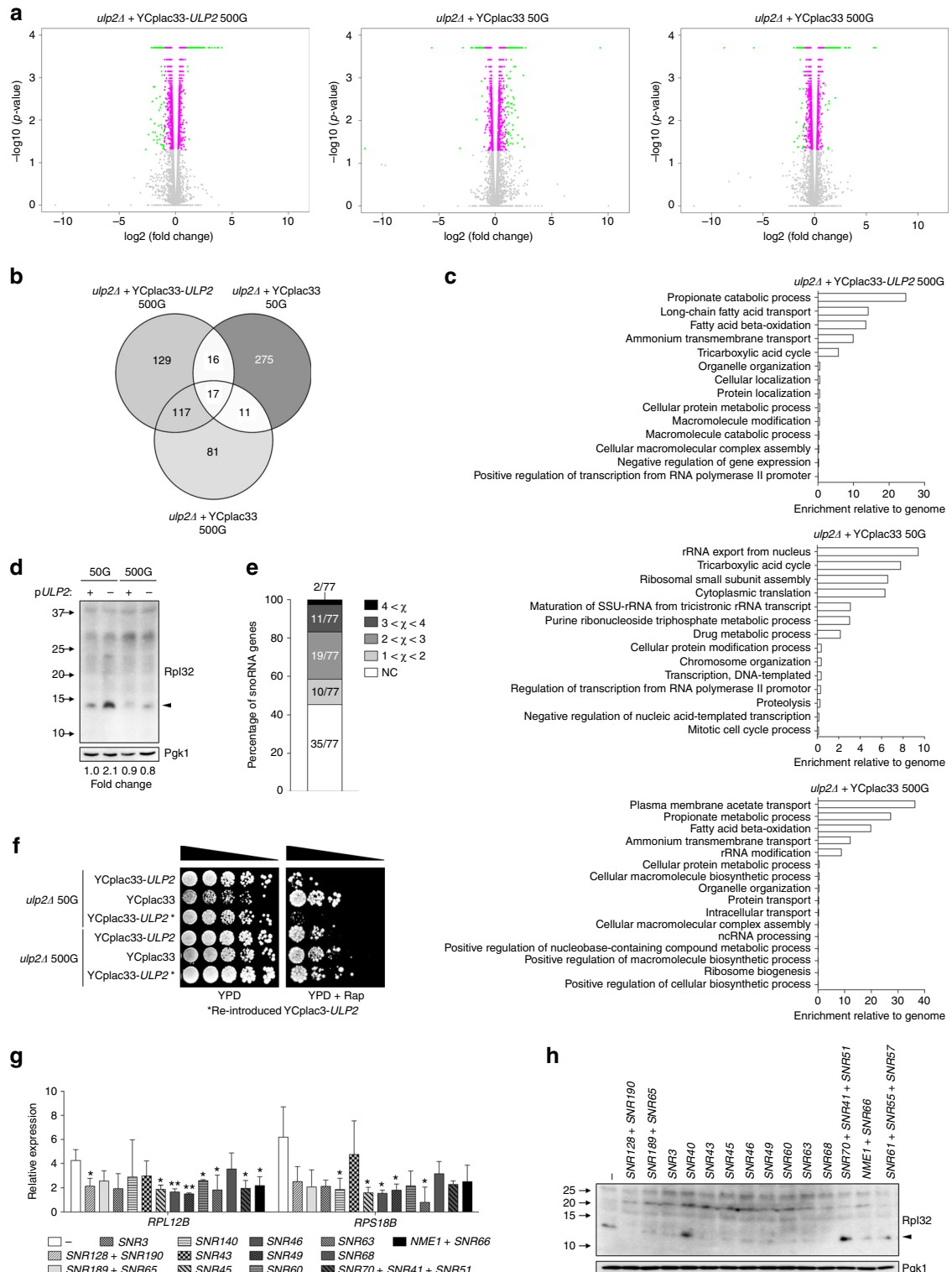

proteins (RPs) and ribosome biogenesis factors were heavily enriched among the genes with strong increases in expression (Fig. 2c (middle) and Supplementary Data 1). Among 136 genes encoding yeast ribosomal subunits, 130 were significantly upregulated in the *ulp2* null mutant. By contrast, high expression of RP genes was no longer seen in *ulp2* null cells after long-term culture at 30 °C in YPD medium (*ulp2Δ* + YCplac33, 500 G; Fig. 2c right). Indeed, these long-term adapted *ulp2Δ* cells had much more overlap in their upregulated transcripts with WT cells than with the nascent *ulp2Δ* strain (Fig. 2b, c). As predicted from

the RNA-seq results, enhanced levels of ribosomal protein Rpl32 in nascent *ulp2Δ* cells were observed, and this returned to control levels after 500 generations. (Fig. 2d).

Earlier genome-wide analysis of SUMO modifications in yeast revealed that SUMO preferentially localizes to genes involved in translation, such as tRNA and RP genes[28], and both RPs and ribosome biogenesis factors undergo SUMOylation[29]. Interestingly, sumoylation of the Rap1 transcription factor cooperates with the target of rapamycin complex 1 (TORC1) kinase pathway to promote transcription of genes encoding RPs[28]. We therefore

**Fig. 2** Analysis of genome-wide gene expression of *ulp2Δ* cells during passaging. **a** Volcano plots displaying differentially expressed genes in the indicated strains as compared to MHY1379 (*ulp2Δ* + YCplac33-*ULP2*) at 50G. The *y*-axis is the mean of the negative logarithm of the *P*-values, and the *x*-axis corresponds the log2 fold change value. Green and magenta dots denote genes significantly changed if the fold change was >2 (|fold changes| > 2) and <2, respectively, and gray dots denote genes with statistically insignificant changes. **b** Venn diagram of genes with significantly increased expression (green dots shown in **a**). **c** Gene Ontology (GO) enrichment analysis of the genes displayed in **b**. Bar diagrams indicate the fold-enrichment of categories of biological process to the genome using GO data from SGD and PANTHER. The genes used in GO biological process are listed in Supplementary Data 1. **d** Immunoblot analysis of the Rpl32 ribosomal protein in extracts from the indicated strains. Anti-Pgk1 blotting was used to verify equal loading. Arrowhead, Rpl32 protein. **e** A percentage graph of snoRNA gene expression that was significantly changed in *ulp2Δ* + YCplac33 at 500G. Genes were binned by fold change (χ). NC, no change. **f** Growth assay of the indicated strains. After spotting cells in fivefold serial dilutions, plates were incubated at 30 °C for 2 (YPD) or 5 days (YPD + Rap). Asterisks indicate reintroduced YCplac33-*ULP2*. Rap, 10 μg/ml rapamycin. **g** qRT-PCR analysis of ribosomal protein genes, *RPL12B* and *RPS18B*, in *ulp2Δ* cells transformed with yeast genomic library plasmids including the indicated snoRNA genes. Expression was measured relative to WT cells (MHY1379) containing empty vector (pGP564). Data were normalized to *ACT1* expression. The error bars indicate the standard deviation (SD) from three independent RNA preparations. Asterisks represent statistically significant differences determined by pairwise comparisons between *ulp2Δ* cells with empty vector (–) and each snoRNA plasmid using a two-tailed Student's *t*-test (*$P < 0.05$; **$P < 0.01$). **h** Immunoblot analysis of the Rpl32 protein in *ulp2Δ* cells containing plasmids with the indicated snoRNA genes, performed as in 2d

tested whether the elevated RP transcripts seen in nascent *ulp2Δ* cells could potentially overcome inhibition of TORC1 by rapamycin, (Fig. 2f). Despite growing more poorly on rich medium alone (YPD), the nascent *ulp2Δ* strain grew better than the matched WT strains on rapamycin (at 50G; top two rows). After long-term passage, all the strains grew better on the drug, although not as robustly as the nascent *ulp2* null strain. These data are consistent with the model that TORC1 and sumoylated Rap1 function in parallel to promote transcription of RP genes[28].

We found that sumoylated Rap1 is a substrate of Ulp2 (Supplementary Fig. 5a). Association of Rap1 with RP genes was not significantly affected by loss of Ulp2 (Supplementary Fig. 5b), consistent with data suggesting sumoylation of Rap1 functions downstream of Rap1 binding to RP promoters[28]. The elevated levels of SUMO-conjugated Rap1 in *ulp2Δ* may therefore help drive RP gene transcription and rapamycin resistance.

**Numerous snoRNAs are elevated in high-passage *ulp2Δ* cells.** Increased levels of several categories of transcripts, such as those for fatty acid beta-oxidation and ammonium transport, were seen in both *ulp2Δ* cells containing a *ULP2* plasmid (WT) and those with an empty vector after 500 generations (Fig. 2c). These might reflect a general response to long-term growth under these laboratory conditions. A much more specific response of long-term adapted *ulp2* null cells was the increase of many snoRNAs, which are critical for rRNA modification (Fig. 2c, right). Of 77 annotated snoRNAs, expression of over half (42) was significantly elevated, compared with WT (Fig. 2e and Supplementary Data 2). The snoRNAs contribute both to the processing of the pre-rRNA into 18s, 5.8s, and 28s rRNAs, and to methylation and pseudouridylation of rRNA[30]. Novel snoRNA functions are also being uncovered, including some important for oncogenesis and different stress responses[31–34]. Notably, a multiple myeloma-associated snoRNA, ACA11, downregulates transcription of RP genes[35].

Although we could not simultaneously upregulate dozens of snoRNAs, as observed in high-passage *ulp2Δ* cells, we could test whether increased levels of specific snoRNA gene(s) could ameliorate any of the *ulp2* mutant defects that characterize low-passage cultures. Yeast 2-μm genomic library plasmids containing one or more snoRNA genes were transformed into WT (MHY1379) cells, and the *URA3*-marked *ULP2* plasmid was then evicted. For these tests, we selected 13 plasmids bearing snoRNAs that displayed comparatively large increases in levels in high-passage *ulp2Δ* cells (Fig. 2e and Supplementary Data 2). None of the high-copy plasmids rescued the growth impairment, aneuploidy, or accumulation of HMW SUMO-conjugated

proteins typical of low-passage *ulp2Δ* cells (Supplementary Fig. 6). However, plasmids containing certain snoRNA genes suppressed the elevated transcripts of the tested *RPL12B* and *RPS18B* RP genes, with eight and five, respectively, of the 13 plasmids suppressing to significant degrees (Fig. 2g). Furthermore, 11 of the 13 plasmids as well as an additional plasmid with three clustered snoRNA genes from ChrXII (see below) reduced excess accumulation of Rpl32 protein upon loss of Ulp2 (Fig. 2h). Whether the identified snoRNAs directly regulate RP expression cannot be inferred from these data, but the inverse correlation in expression between a cohort of snoRNAs and RP genes is supported. Inasmuch as snoRNAs are essential for ribosome formation, it is possible that enhanced ribosome biogenesis under these snoRNA overexpression conditions partially relieves the requirement for higher RP gene expression.

**Ulp2 modulates SUMO modification at RP and snoRNA genes.** Past studies have implicated mammalian SUMO proteases in numerous transcriptional processes[16], but there was no evidence that yeast Ulp2 is directly involved in transcription. To determine whether Ulp2 is recruited to RP or snoRNA genes, we carried out chromatin immunoprecipitation (ChIP) assays using cells with the chromosomal *ULP2* gene tagged with a sequence encoding a Flag epitope. We used the Rap1 transcription factor as a positive control as it is known to associate with the RP genes and some snoRNA genes[28]. The ChIP analysis indicated that Ulp2 associated with the promoter and coding sequence (CDS) of both RP and snoRNA genes (Fig. 3a); the protein deacetylase Hst3 and the transcriptional repressor Nrg2, whose genes were significantly downregulated in *ulp2Δ* cells after extensive passaging, were not associated with these genes. These data suggest that Ulp2 could be directly involved in transcription of these gene classes.

Association of the Ulp2 SUMO protease with chromatin sites would be predicted to decrease local protein sumoylation, as occurs at the RP genes[28]. We compared levels of yeast SUMO (Smt3) at the RP and snoRNA genes in WT and *ulp2Δ* cells (Fig. 3b). His6-tagged Smt3 was used for the ChIP analysis. Loss of Ulp2 was associated with increased levels of SUMO/Smt3 in both the promoter and CDS regions at most of the tested loci.

Taken together, our data reveal that Ulp2 associates with RP and snoRNA loci where it limits local accumulation of SUMO (presumably SUMO-conjugated chromatin proteins such as Rap1 or histones). This suggests that Ulp2 has a direct role in the expression of RP and snoRNA genes.

**Increased Ccr4 in *ulp2Δ* limits expression of snoRNA and rRNA.** Many rDNA silencing factors are targets of sumoylation,

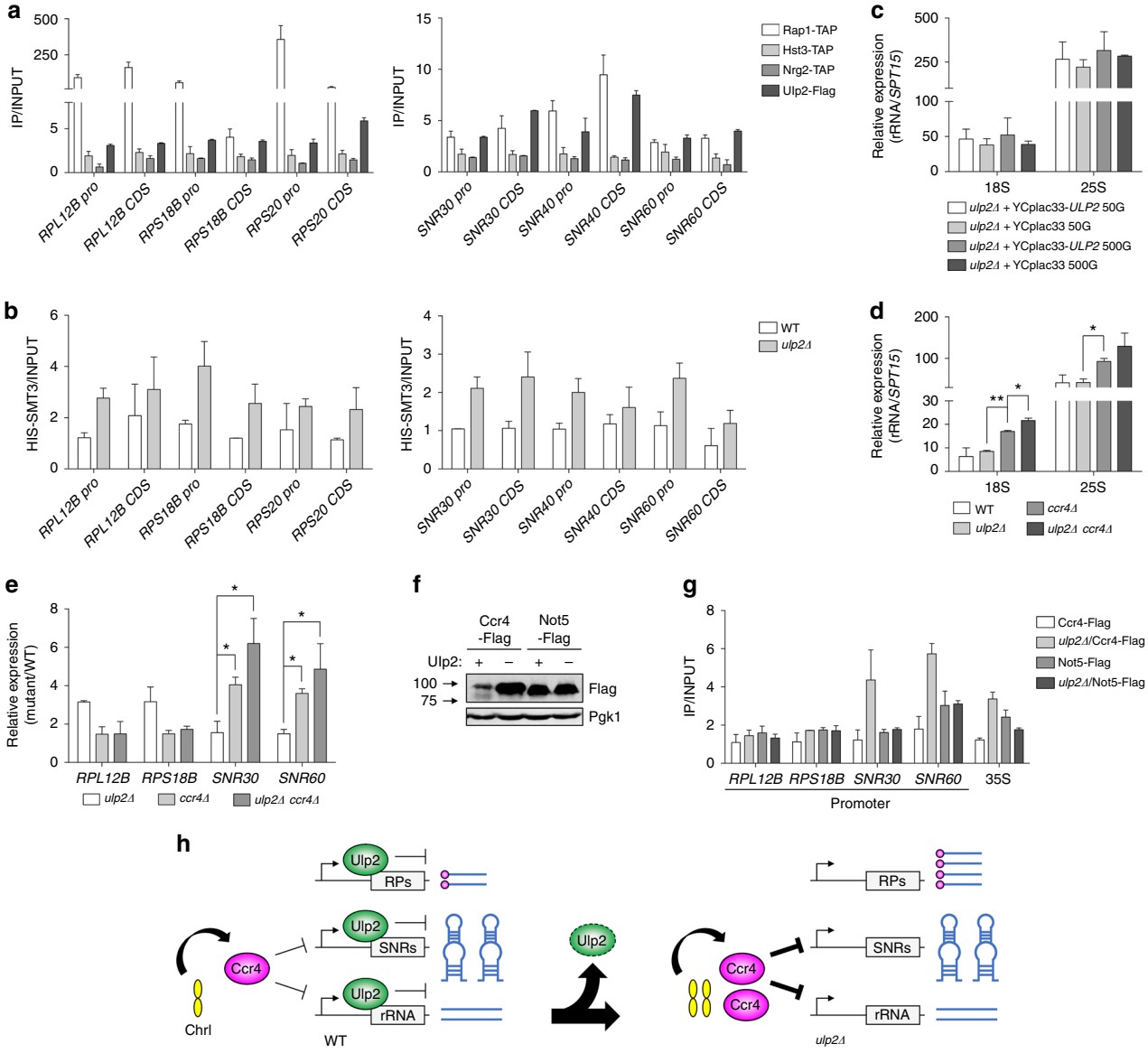

**Fig. 3** Ulp2 localizes to ribosomal protein and snoRNA genes and affects local sumoylation. **a** Chromatin immunoprecipitation (ChIP) analysis using IgG-Sepharose or anti-Flag agarose beads and strains expressing TAP-tagged Rap1, Hst3, or Nrg2 or Flag-tagged Ulp2. Real-time PCR signals at promoter (Pro) and coding sequence (CDS) regions of indicated genes were quantitated and normalized to an internal background control and the input DNA. The error bars indicate the SD calculated from two independent chromatin preparations. **b** Occupancy of Smt3 at the indicated genes was determined in the WT and *ulp2Δ* strains expressing His-tagged Smt3 by ChIP using Ni$^{+2}$-NTA resin and analyzed as shown in **a**. The error bars indicate the SD from two independent preparations. **c**, **d** qRT-PCR analysis of 18s and 25s rRNA in the indicated strains done as in Fig. 2g. Data were normalized to *SPT15* expression. The error bars indicate the SD from two independent RNA samples, and asterisks indicate statistically significant differences (*$P < 0.05$; **$P < 0.01$, two-tailed Student's *t*-test). **e** Gene expression analysis of ribosomal protein genes *RPL12B* and *RPS18B* and snoRNA genes *SNR30* or *SNR60* by qRT-PCR assay, performed as in Fig. 2g. Expression was measured relative to WT cells (MHY500). Data were normalized to *SPT15* expression. The error bars represent the SD from two independent RNAs, and asterisks indicate statistically significant differences (*$P < 0.05$, two-tailed Student's *t*-test). **f** Immunoblot analysis of Ccr4-Flag or Not5-Flag in the indicated *ulp2Δ* strains. Anti-Pgk1 blotting was used to verify equal loading. **g** ChIP assay using anti-Flag agarose beads in the indicated strains as shown in **a**. The error bars indicate the SD from two independent experiments. **h** Proposed model of transcription of genes required for ribosome biogenesis in *ulp2Δ* cells. RPs, SNRs, and rRNA indicate genes of ribosomal proteins, snoRNAs and ribosomal RNA, respectively. Normally, Ulp2 represses transcription of RP, snoRNA, and rRNA genes, and Ccr4 protein, encoded on ChrI, represses expression of snoRNAs and rRNA but not RPs. Loss of Ulp2 leads to ChrI disomy, and the increased level of ChrI-encoded Ccr4 counters the loss of Ulp2-dependent repression of transcription of snoRNA and rRNA, but not RP, genes

and at least the Net1, Tof2, Cdc14 and Fob1 nucleolar proteins are substrates of Ulp2[36,37]. Ulp2 localizes to rDNA loci, and Ulp2 mutations strongly impair association of Net1, Tof2 and Fob1 with rDNA[36], suggesting Ulp2 generally promotes rDNA silencing. We therefore assumed that rRNA levels would be altered in *ulp2Δ* cells undergoing laboratory evolution; however, rRNA levels were unchanged in cells passaged for 50 or 500 generations (Fig. 3c).

We previously reported that ChrI aneuploidy in *ulp2Δ* cells is suppressed by increased dosage of two ChrI genes, one encoding Ccr4, a subunit of the multifunctional Ccr4–Not complex, and the other Cln3, a G$_1$-phase cyclin[18]. Therefore, raising either Ccr4

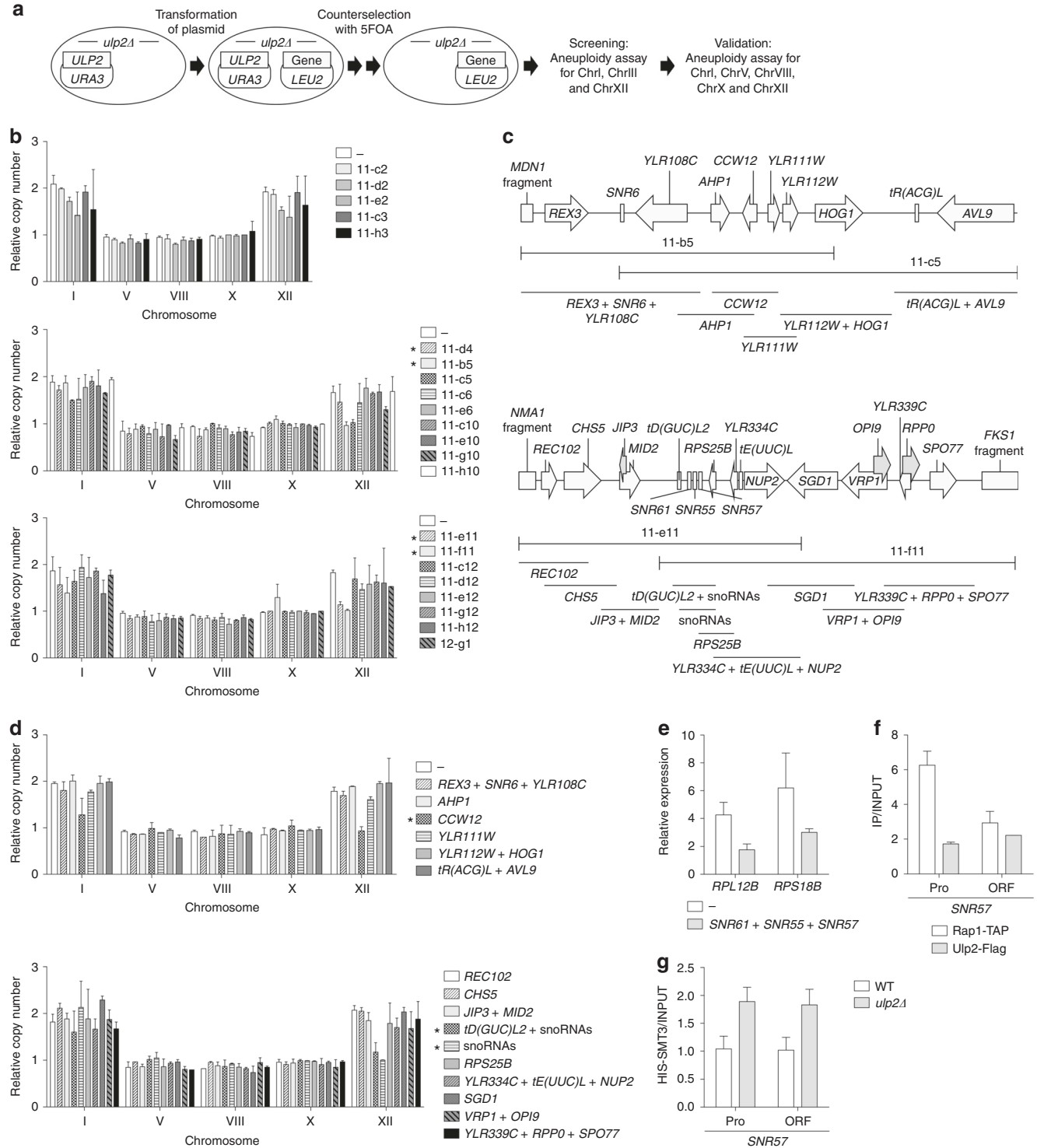

or Cln3 levels is sufficient to prevent the ChrI duplication. Notably, the Ccr4-Not complex limits RNA Pol I binding to rDNA loci and inhibits rRNA synthesis[38]. Moreover, a wide variety of snoRNAs are over-expressed in mutants of the Ccr4-Not complex[39,40].

We confirmed that lack of Ccr4 resulted in higher expression of rRNA (18s and 25s), *SNR30*, and *SNR60* (Fig. 3d, e); RP transcript levels were unaffected. The *ulp2Δ* mutation by itself did not alter expression of these genes, but in a *ulp2Δ ccr4Δ* strain, these RNAs, and not RP transcripts, appeared to be modestly

increased relative to *ccr4Δ*. Ccr4 protein levels rose sharply as a result of the ChrI disomy in nascent *ulp2Δ* cells (Fig. 3f); this increase was accompanied by a specific enrichment of Ccr4, but not the Not5 subunit of the Ccr4-Not complex, at snoRNA gene promoters as well as the 35s gene (Fig. 3g).

Taken together, our data and previous results suggest that Ulp2 may generally limit transcription of the RP, snoRNA and rRNA genes, whereas Ccr4 blocks the expression of snoRNA and rRNA genes, but not of RP genes (Fig. 3h, left). Loss of Ulp2 leads to increased Ccr4 as a result of the ChrI disomy, which in turn limits

**Fig. 4** Identification of genes on ChrXII suppressing *ulp2Δ* aneuploidy. **a** Strategy used to identify specific regions of ChrXII that suppress *ulp2Δ* aneuploidy. A *ulp2Δ*/YCplac33-*ULP2* strain (MHY1379) was transformed with the indicated pGP564 (*LEU2*)-based ChrXII library plasmids. Cells were struck on SD–Leu + FOA twice in succession to evict YCplac33-*ULP2*. After the putative candidate plasmids were selected by qPCR-based aneuploidy assays of ChrI, ChrIII, and ChrXII, the copy number of ChrI, ChrV, ChrVIII, ChrX, and ChrXII was determined in *ulp2Δ* cells bearing the selected plasmids (**b**) or plasmids including specific ChrXII fragments (**d**). **b** qPCR ploidy assays of *ulp2Δ* cells with the high-copy plasmids from the first round of selection. Copy number of the five tested chromosomes was normalized to the euploid control strain (MHY1379 + pGP564) and compared with *ulp2Δ* cells expressing empty pGP564 vector. Error bars indicate the SD calculated from two independent genomic DNA preparations. Asterisks indicate plasmids showing a reduction of ChrXII disomy. **c** Schematic diagrams of plasmids pGP564-11-b5, pGP564-11-c5, pGP564-11-e11 and pGP564-11-f11. Locations of ORFs (arrows or boxes) and gene names are indicated. Subclones used in **d** are shown as bars below the ChrXII regions. Full-length genes contained within subclones are listed below the bars. **d** qPCR ploidy assays of *ulp2Δ* cells with the high-copy plasmids shown in **c**. Chromosome copy number was determined as in **b**. Error bars indicate the SD calculated from two independent experiments. Asterisks denote plasmids showing a loss of ChrXII disomy. **e** qRT-PCR analysis of *RPL12B* and *RPS18B* in *ulp2Δ* cells containing a plasmid with snoRNAs (11-e11) as in Fig. 2g. The data of (–) is identical with the (–) signal shown in Fig. 2g. The error bars represent the SD from three experiments. **f** ChIP analysis of Rap1-TAP and Ulp2-Flag strains as in Fig. 2a. The error bars indicate the SD from analyses of two independent chromatin preparations. **g** SUMO levels measured by ChIP assay in the WT and *ulp2Δ* strains expressing His-tagged Smt3 (SUMO) as in Fig. 2b. The error bars indicate the SD from two independent samples

the transcription of snoRNA and rRNA genes (Fig. 3h, right). The net effect is a specific increase in RP expression in incipient *ulp2Δ* cells.

**Exogenous ChrXII snoRNAs suppress the ChrXII *ulp2Δ* disomy**. Next, we screened for genes on ChrXII that could prevent the ChrXII disomy upon *ULP2* deletion (Fig. 4a). Each of 135 plasmids containing individual ChrXII fragments[41] was transformed into *ulp2Δ* cells carrying a *URA3*-marked *ULP2* plasmid (MHY1379), and the *ULP2* plasmid was then evicted. Ratios of ChrI and ChrXII to ChrIII, a chromosome that is not duplicated in *ulp2Δ* cells, were analyzed by a qPCR-based ploidy assay (Supplementary Fig. 7).

Transformants with 22 different library plasmids reduced the ratio of ChrI/ChrIII or ChrXII/ChrIII from ∼2 to 1, suggesting possible suppression of *ulp2Δ* aneuploidy. We repeated the ploidy assay on five chromosomes with the candidate plasmids in nascent *ulp2Δ* cells (Fig. 4b). Four plasmids covering two separate regions of ChrXII continued to suppress disomy of ChrXII but not of ChrI, (Fig. 4b, c). They did not affect the slow growth and accumulation of polySUMO conjugates (Supplementary Fig. 8). To determine the genes within these clones required for suppression of the *ulp2Δ* aneuploidy, 15 DNA fragments were individually cloned into library vector pGP564 (Fig. 4c), and their effect on ploidy was tested (Fig. 4d). Extra copies of *CCW12* or a cluster of snoRNA genes (*SNR61*, *SNR55*, and S*NR57*) specifically prevented the development of the ChrXII disomy. Ccw12 is a mannoprotein important for cell wall integrity, and its cellular level is tightly regulated[42–44]. The possible contribution of Ccw12 to *ulp2Δ* aneuploidy was not pursued further here.

The three snoRNA genes in the cloned cluster were all C/D box snoRNAs. Besides suppressing the ChrXII disomy, high-copy expression of the cluster suppressed the excess expression of RP genes *RPL12B* and *RPS18B* normally observed in nascent *ulp2* mutant cells (Fig. 4e); the much higher dosage of these three snRNA genes compared to ChrXII disomy presumably accounts for the suppression of RP transcripts only in the former case. As predicted from Fig. 3a, Ulp2 was directly recruited to both promoter and coding sequences of *SNR57* (Fig. 4f), and SUMO levels at these sites increased in *ulp2Δ* cells (Fig. 4g), suggesting that Ulp2 is involved in the transcription of *SNR57* as well as other snoRNA genes. These data support the hypothesis that enhanced levels of snoRNAs help circumvent the cellular stress caused by loss of Ulp2, first through a ChrXII disomy that raises the dosage of these C/D box snoRNAs, and then after many generations, through a broad increase in expression of many snoRNAs, which is linked to suppression of excess RP transcripts.

**Table 1 Mutations in the original *ulp2Δ* strains passaged for 500 cell generations**

| Strains | Chr | Position | Reference DNA sequence(s) | Altered DNA sequence(s) | Gene |
|---|---|---|---|---|---|
| *ulp2Δ* + YCplac33-*ULP2* (WT) | II | 809923 | AA | GG | – |
| | II | 809938 | G | A | – |
| | II | 810286 | A | T | – |
| | VII | 873759 | A | G | *BUB1* |
| *ulp2Δ* + YCplac33 | III | 14720 | G | A | – |
| | IV | 337677 | A | T | *UBC9* |
| | IX | 439868 | A | G | – |
| | XV | 279630 | T | C | *IFM1* |
| *ulp2Δ* + YCplac33-*ULP2* (reintroduced) | II | 720581 | G | A | |
| | | *SPO23* | X | 410712 | C |
| | T | *MAD3* | | | |
| | XII | 598800 | TAG | TAAG | – |
| | XII | 598804 | A | T | – |
| XV | | 380372 | G | C | *BUB3* |

**High-passage *ulp2Δ* cells acquire adaptive mutations**. Cellular mechanisms for adaptation to stress include gene mutation, aneuploidy, and transcriptional or translational changes, depending on the nature and length of the stress[3,4]. In a previous study, aneuploidies developed under heat or high pH stress were superseded over longer periods by gene mutations and refinements of gene expression presumed to carry lower fitness costs than aneuploidy[7]. We asked if aneuploid *ulp2Δ* yeast undergo an analogous evolutionary process using whole genome DNA sequencing (DNA-seq) to identify potential gene mutations arising during repeated cell passaging (Table 1). Mutations were detected in both coding and noncoding DNA sequences; we focused on those predicted to cause changes to protein sequences.

Interestingly, *BUB1*, *MAD3*, and *BUB3* genes, which are involved in the spindle checkpoint of the cell cycle[45], and *SPO23* gene, encoding a protein that interacts with meiosis-specific protein Spo1[46], carried missense mutations in evolved *ulp2Δ* cells that either continuously carried or had reintroduced the *ULP2* plasmid. Although these *ulp2Δ* strains had a *CEN* plasmid with the WT *ULP2* gene, plasmid instability might reduce Ulp2 expression in these cultures[17]. This could cause cell-cycle arrest in response to checkpoint activation[47], a defect that may be overcome by mutations in the checkpoint genes. The adaptive significance of the Spo23 mutation is unknown.

In the *ulp2Δ* strain that had been passaged for ∼500 generations without *ULP2* plasmid (*ulp2* null cells), two missense

mutations were found (Table 1). One was in *IFM1*, encoding mitochondrial translation initiation factor 2; this was not investigated further. The other mutation was in *UBC9*, which codes for the SUMO-conjugating E2 enzyme. The predicted K27N substitution is in an N-terminal domain that contains binding sites for the E1 and E3 enzymes[48]. Accumulation of polySUMO conjugates in these late-passage cells had attenuated as well (Fig. 1d).

**Analysis of *ulp2Δ* descendants after long-term passaging**. To assess the generality of the altered mutant phenotype of evolved *ulp2Δ* cells observed in Figs. 1 and 2, we grew ten separate *ulp2Δ* lines from a single nascent *ulp2Δ* colony and evaluated them after 250 and 500 generations. ChrXII disomy, but not that of ChrI, had disappeared by 250 generations in all the evolved lines, as had their growth defects (Fig. 5a, b), consistent with Fig. 1b, c and our previous report[18]. After an additional 250 generations, the disomy of ChrI was also eliminated. Analysis by FACS showed the recovery from the initial *ulp2Δ* cell-cycle defect at both 250 and 500 generations (Fig. 5c). Also, after both 250 and 500 generations, all the *ulp2Δ* strains showed resistance to high temperature and oxidative stress (Supplementary Fig. 9). Consistent with RNA-seq data obtained from *ulp2Δ* at 500 generations in Fig. 2, all ten of the high-passaged *ulp2Δ* strains showed reversion of the *RPL12B* and *RPS18B* transcripts to normal levels along with elevated transcripts of snoRNAs, *SNR30* and *SNR60* (Fig. 5d). Thus, all *ulp2Δ* strains eventually move away from aneuploidy and adopt alternative adaptive strategies that enhance fitness.

As noted earlier, after 500 generations of continual passaging, the original *ulp2Δ* strain tested no longer accumulated excess SUMO conjugates (Fig. 1d). We also measured SUMO conjugate levels by anti-SUMO immunoblotting in the ten separately evolved *ulp2Δ* strains after 250 and 500 generations (Fig. 5e). PolySUMO-conjugated proteins significantly declined in five of the strains after 250 generations (strains 3, 4, 8, 9, and 10). After 500 generations, conjugate levels had also declined in strains 1 and 7.

Hence, all of the independently evolved *ulp2Δ* strains eventually reestablished euploidy and near-normal cell growth and cell-cycle progression, but even after 500 generations, several of the strains still accumulated excess HMW SUMO conjugates, suggesting alternative long-term mechanisms for adapting to Ulp2 loss.

**Point mutations in the SUMO E1 enzyme in evolved *ulp2Δ***. In the original evolved *ulp2Δ* strain, a mutation in the *UBC9* gene had been uncovered (Table 1), presumably because reduced E2 activity suppresses accumulation of polySUMO conjugates. No *UBC9* mutations were found in the ten new *ulp2Δ* strains grown without *ULP2* for 250 and 500 generations (Table 2). Since most of the evolved *ulp2Δ* strains showed reduced accumulation of polySUMO conjugates (Fig. 5e), we checked for mutations in all other genes known to be required for normal SUMO conjugation in vegetative cells, specifically those encoding the E1 (Aos1/Uba2 dimer), E3s (Siz1, Siz2 and Mms21), and SUMO protease Ulp1.

Interestingly, we found several missense mutations in *UBA2*, which encodes the catalytic E1 subunit. All ten strains shared a mutation in the *UBA2* ORF, causing a C162S change in the encoded protein (Fig. 5f). This mutation was probably acquired during the original 5-FOA counter-selection to evict the covering *ULP2* plasmid because all ten *ulp2Δ* strains originated from the same colony after growth for ~30 generations on 5-FOA. Additional *UBA2* missense mutations were observed in six of the evolved *ulp2Δ* strains (Fig. 5f). All these *ulp2Δ* strains expressed a doubly mutated Uba2 and suppressed accumulation

of polySUMO conjugates (Fig. 5e). Strain 8, which also suppressed polySUMO-conjugate levels yet had no additional Uba2 mutations, picked up a mutation in Aos1 (E346V). Interestingly, while the uba2-C162S mutation was insufficient to suppress polySUMO-conjugate levels in strains 1 and 7 at 250 generations, the additional Uba2 mutations seen at 500 generations were associated with a strong decrease in conjugate levels. Conversely, three evolved *ulp2Δ* strains (2, 5, and 6) had no detectable changes in the tested SUMO pathway components, consistent with their continued high levels of polySUMO conjugates (Fig. 5e).

The Uba2 missense mutations occurred throughout the protein (Fig. 5f). Only C162S is located within the domain bearing the catalytic Cys173 residue. The other mutations occur in the adenylation domain (AD) or the ubiquitin-like domain (UFD) required for recruiting E2[49]. When provided as the only source of Uba2 in otherwise WT cells, all the Uba2 point mutants, including the $Uba2^{C162S}$ single mutant and the six Uba2 double point mutants, reduced SUMO conjugation (Fig. 5g) without impairing cell growth (Supplementary Fig. 10). Cells expressing $Aos1^{E346V}$ continued to accumulate conjugates, but the *aos1-E346V uba2-C162S* double mutant did not.

Taken together, our results suggest that long-term loss of Ulp2 SUMO protease activity favors countervailing partial loss-of-function mutations in genes encoding SUMO ligation pathway enzymes, particularly the essential Ubc9 (E2) and Uba2/Aos1 (E1) enzymes.

***UBC9* and *UBA2* mutations prevent aneuploidy in nascent *ulp2Δ***. To test the hypothesis that *ulp2Δ* aneuploidy could be directly suppressed by these balancing mutations in SUMO-ligating enzymes, we measured chromosome copy number, as well as cell growth, cell-cycle progression, and SUMO conjugates, in nascent *ulp2Δ* cells that carried *ubc9*, *uba2*, or *aos1* missense mutations (Fig. 6a). $Ubc9^{K27N}$ expression efficiently suppressed the development of aneuploidy and the other mutant traits in low-passage *ulp2Δ* cells (Fig. 6b–e). Notably, even though provision of the single point mutant $Uba2^{C162S}$ did not suppress the growth and cell-cycle defects of nascent *ulp2Δ* cells, it slightly reduced levels of HMW SUMO-conjugated proteins and prevented the development of ChrXII (but not ChrI) disomy (Fig. 6f–i). Aneuploidy and the other defects of cells newly depleted of Ulp2 were also prevented by all tested *uba2* double point mutants and the $uba2^{C162S}$ $aos1^{E346V}$ double mutant (Fig. 6f–m). These suppression results with nascent *ulp2* null cells strongly support the contention that the E1 and E2 missense mutations isolated through long-term adaptation to Ulp2 loss are directly responsible for the recovery from aneuploidy and the suppression of the growth and stress resistance defects.

## Discussion

Following an acute loss of the Ulp2 SUMO protease, yeast respond by developing a specific two-chromosome aneuploidy[18]. While this rapid adaptive response helps stave off inviability, the mutant has many severe defects. The present study demonstrates that upon extended laboratory evolution, the extra chromosomes are lost in step-wise fashion. Remarkably, despite the complete absence of Ulp2, these evolved euploid cells acquire nearly normal growth and cell-cycle characteristics. Homeostasis appears to be restored in most cases by a combination of two mechanisms (Fig. 7). First, partial loss-of-function mutations to the essential E1 or E2 SUMO pathway enzymes reduce the excess accumulation of polySUMO-modified proteins caused by loss of Ulp2. Second, regulation of ribosome biogenesis, which is tightly controlled by the SUMO system, is altered such that overexpression

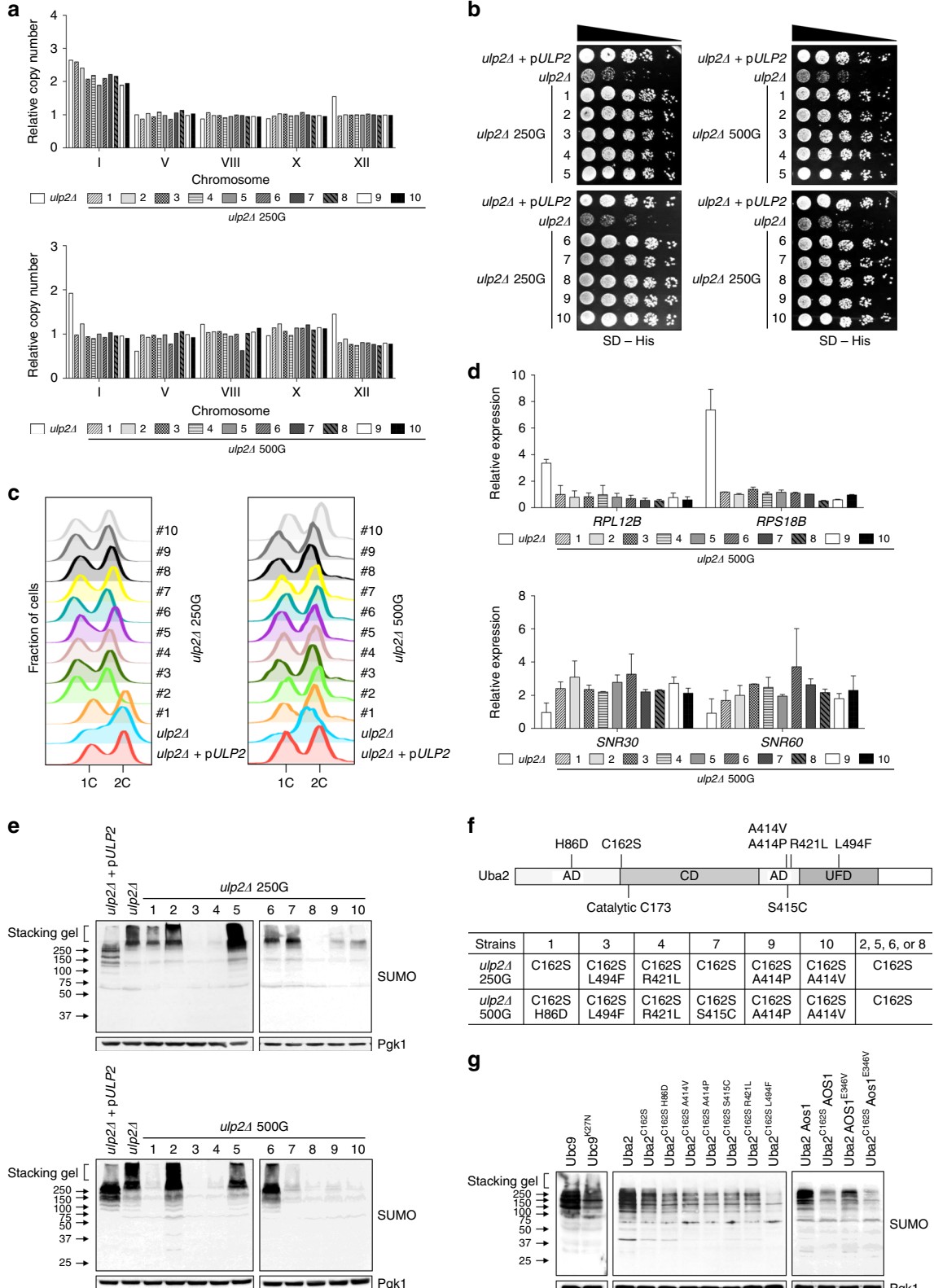

**Fig. 5** *UBA2* and *AOS1* genes are mutated in high-passage *ulp2Δ* cells. **a–e** qPCR ploidy assays (**a**), growth assays (**b**), flow cytometry analysis (**c**), qRT-PCR analysis of ribosomal protein genes *RPL12B* and *RPS18B* and snoRNA genes *SNR30* or *SNR60* (**d**), and immunoblot analysis of SUMO-conjugated proteins (**e**) were performed for ten independently evolved *ulp2Δ* lines at 250 or 500 generations. Gene expression was measured relative to WT cells (MHY1379) and data were normalized to *SPT15* expression in **d**. The error bars indicate the SD from two experiments. **f** Schematic of Uba2 protein and list of mutations in *ulp2Δ* 250G or 500G strains. AD, CD, and UFD indicate adenylation, Cys and ubiquitin-like fold domains, respectively. Positions of the catalytic Cys residue and mutations identified in Table 2 are noted. **g** *ubc9Δ*, *uba2Δ*, or *uba2Δ aos1Δ* strains carrying a YCplac33 plasmid with *UBC9*, *UBA2*, or *UBA2 AOS1* were transformed with a pRS315 plasmid expressing WT or the indicated mutant proteins. After shuffling out the *URA3*-marked plasmids, anti-SUMO immunoblot analysis of cell extracts was performed

**Table 2 Mutations of SUMO enzymes caused by Ulp2 loss during multiple cell generations**

| Strains | No. | AOS1 | UBA2 | UBC9 | SIZ1/2 | MMS21 | ULP1 |
|---|---|---|---|---|---|---|---|
| ulp2Δ 500G | 1 | NC[a] | 484 (T to A), 256 (C to G) | NC | NC | NC | NC |
| | 2 | NC | 484 (T to A) | NC | NC | NC | NC |
| | 3 | NC | 484 (T to A), 1482 (G to T) | NC | NC | NC | NC |
| | 4 | NC | 484 (T to A), 1262 (G to T) | NC | NC | NC | NC |
| | 5 | NC | 484 (T to A) | NC | NC | NC | NC |
| | 6 | NC | 484 (T to A) | NC | NC | NC | NC |
| | 7 | NC | 484 (T to A), 1244 (G to T) | NC | NC | NC | NC |
| | 8 | 1037 (A to T) | 484 (T to A) | NC | NC | NC | NC |
| | 9 | NC | 484 (T to A), 1240 (G to C) | NC | NC | NC | NC |
| | 10 | NC | 484 (T to A), 1241 (C to T) | NC | NC | NC | NC |
| ulp2Δ 250G | 1 | NA[b] | 484 (T to A) | NA | NA | NA | NA |
| | 2 | NA | 484 (T to A) | NA | NA | NA | NA |
| | 3 | NA | 484 (T to A), 1482 (G to T) | NA | NA | NA | NA |
| | 4 | NA | 484 (T to A), 1262 (G to T) | NA | NA | NA | NA |
| | 5 | NA | 484 (T to A) | NA | NA | NA | NA |
| | 6 | NA | 484 (T to A) | NA | NA | NA | NA |
| | 7 | NA | 484 (T to A) | NA | NA | NA | NA |
| | 8 | 1037 (A to T) | 484 (T to A) | NA | NA | NA | NA |
| | 9 | NA | 484 (T to A), 1240 (G to C) | NA | NA | NA | NA |
| | 10 | NA | 484 (T to A), 1241 (C to T) | NA | NA | NA | NA |
| ulp2Δ | – | NA | NC | NA | NA | NA | NA |

[a]NC (no change)
[b]NA (no analysis)
ATG: +1

of RP genes is reversed and snoRNAs are broadly upregulated, the latter likely through attenuated Ccr4-dependent transcriptional control. These data demonstrate a striking flexibility of cells in their response to loss of an essential gene: a specific but transient aneuploid state followed by distinct evolutionary trajectories capable of resolving the chromosome imbalances and restoring normal growth and stress resistance. Such interdependent adaptive mechanisms likely have relevance to other long-term cellular stress responses, including the evolution of tumor cells and aging.

Yona et al.[7] previously found that disomies induced by culturing yeast at either high temperature or high pH were eventually resolved even in the continued presence of the stress. Resolution of aneuploidy following the loss of an essential gene has not be reported previously to our knowledge. Our data indicate that different aneuploid ulp2Δ lines undergo variable but to some extent predictable adaptive evolutionary changes to compensate for the missing gene in ways that allows normal chromosome numbers to be restored. Predictable evolutionary trajectories were previously observed in yeast strains in response to the removal of an important cell polarity factor, although this did not involve an initial aneuploid state[50].

The different genetic alterations of separately evolved aneuploid ulp2Δ lines is in some ways reminiscent of the genetic heterogeneity that characterizes human tumors[51]. Over 90% of solid tumors are aneuploid, but extensive intratumor variability is observed. In some tumors, distinct mutations affecting the same tumor driver genes were found, suggesting convergent evolution leading to neoplastic growth and metastasis. In the transiently aneuploid ulp2Δ lines in our study, missense mutations in three different SUMO-conjugation factors were identified, and for Uba2, multiple independent missense mutations, as these cells evolved back to a fast-growing euploid state.

A natural experiment in the loss of the ULP2 gene might have occurred in microsporidial species. Microsporidia are intracellular parasites related to fungi that have undergone drastic genome reductions[52]. In Encephalitozoon spp., which have only ~2000 genes, no clear ULP2 ortholog is found, although at least a rudimentary SUMO ligation system still appears to exist based on

available genome sequences. Different microsporidial species might have independently evolved ways of maintaining a balance between SUMO conjugation and deconjugation as components of the SUMO system were lost or reduced.

It should be possible in many cases to trace the complex pathways of long-term genetic change occurring in response to loss or inhibition of cellular enzymes, as demonstrated in the current study. The ability of aneuploidy to serve as a suboptimal but rapid response to cellular stress, is now well documented; However, long-term responses to aneuploidy have been only minimally investigated[7]. These responses may not only be relevant to tumorigenesis, as noted above, but to aging and drug resistance. For example, we found that yeast ulp2 mutants are unusually resistant to rapamycin. Rapamycin treatment boosts longevity in many species, and mTORC1 inhibitors are now being considered for use in humans[53]. Our data suggest that genetic background and tissue heterogeneity may have major effects on the efficacy of such treatments.

## Methods

**Yeast strains**. Yeast strains used in this study are listed in Supplementary Table 1. Standard techniques were used for strain construction. The mutant ulp2Δ ubc9 strain was made by replacing the WT UBC9 ORF in MHY1328 (ulp2Δ/ULP2 diploid) with a TRP1 marker; the resulting strain was transformed with YCplac33-UBC9, sporulated and dissected, and the doubly deleted haploid was isolated. After transforming in the pRS315-ubc9 plasmid, YCplac33-UBC9 was evicted by two consecutive streakings on 5-FOA. The ulp2Δ uba2 and ulp2Δ uba2 aos1 strains were made in analogous fashion except the UBA2 gene was replaced by a kanMX6 module and AOS1 by a natMX4 cassette. To generate MHY9393, MHY10229 and MH10230 strains, the C-terminal insertion cassettes of CCR4–6xGly-3xFlag, NOT5–6xGly-3xFlag and RAP1–6xGly-3xFlag were constructed by PCR amplification from pFA6a-6xGly-3xFlag::KanMX6[54] and transformed into MHY1379 cells (ulp2Δ + YCplac33-ULP2 haploid).

To create yeast strains newly null for ULP2, MHY1379 cells with or without a LEU2-marked plasmid carrying a gene(s) of interest were streaked on 5-FOA or 5-FOA–Leu plates twice in succession to evict the ULP2 cover plasmid. All strains were verified by PCR and/or immunoblot analysis.

**Yeast growth conditions**. Cells were grown at 30 °C in YPD or synthetic defined (SD) minimal medium with the appropriate supplements. For growth analysis on plates, cultures in exponential growth were normalized to an $A_{600}$ of 0.1, which

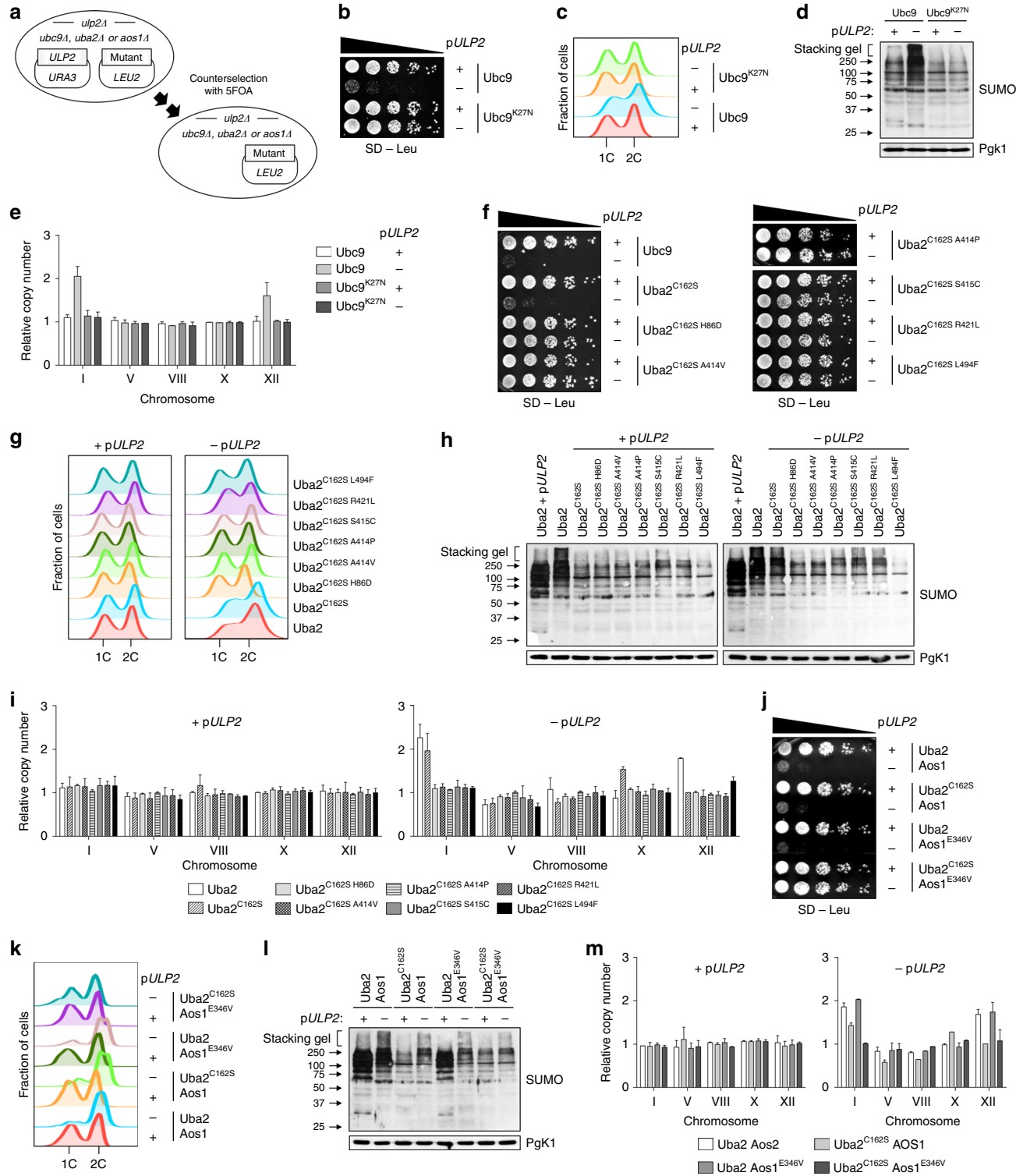

**Fig. 6** Mutations in Ubc9, Uba2, and Aos1 suppress *ulp2Δ* aneuploidy and other defects. **a** Plasmid eviction strategy to test *ubc9*, *uba2*, and *uba2 aos1* mutants that were also *ulp2Δ*. **b**, **f**, **j** Growth assays of the indicated strains. After spotting cells in fivefold serial dilutions, SD–Leu plates were incubated for 2 days at 30 °C. **c**, **g**, **k** Flow cytometry analysis of DNA content of the indicated strains. **d**, **h**, **l** Immunoblot analysis of SUMO conjugates in whole-cell lysates. **e**, **i**, **m** qPCR ploidy assays. Error bars indicate the SD calculated from two independent experiments

were subject to fivefold serial dilutions. Dilution series were spotted onto appropriate media, and the plates were incubated at 30 or 37 °C for 2–5 days. To induce Ulp2 degradation via the AID* tag and *ULP2* mRNA turnover via the neomycin-inducible hammerhead ribozyme (HHR) in the 3′ UTR of *ULP2*, cells were incubated in YPD with 500 μM indole-3-acetic acid (IAA) and 100 μg/ml of neomycin[55,56] and then diluted daily 1:120 into fresh medium containing the drugs for 16 days. For laboratory evolution experiments, cells were grown to stationary

phase in YPD and then diluted 1:120 into fresh YPD medium, as described previously[7]. This process was repeated daily (6.9 generations per day).

**Plasmids**. Plasmids used in this study are listed in Supplementary Table 2. To create high-copy plasmids with ChrXII subfragments from the 11-b5, 11-c5, 11-e11, or 11-f11 genomic library clones, the ORFs with 0.6–1 kb of sequence upstream and downstream of the ORF were PCR-amplified from the genomic

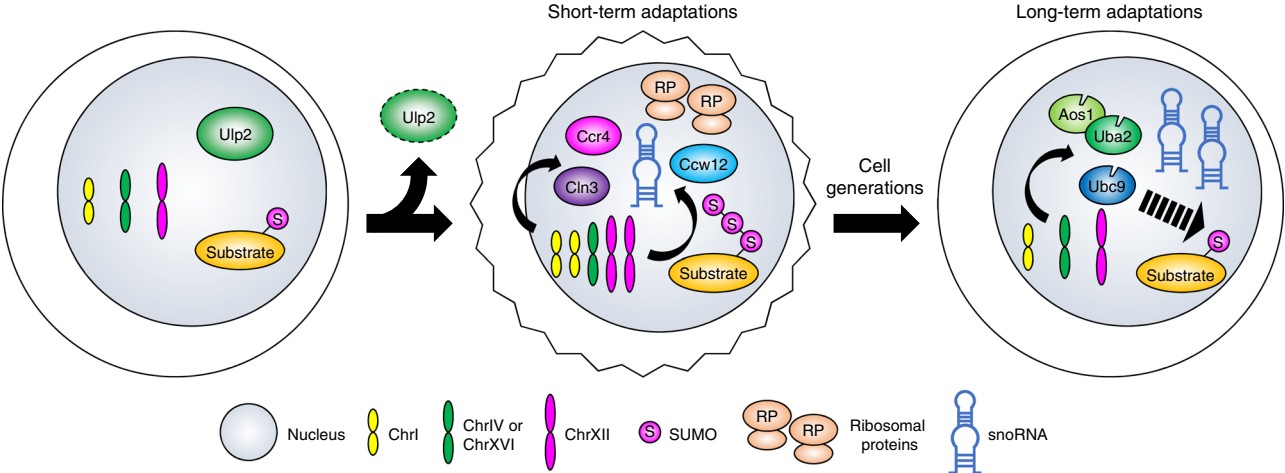

**Fig. 7** Short-term and long-term adaptive mechanisms caused by loss of Ulp2. Proposed model showing that yeast have short-term and long-term adaptive mechanisms that allow survival of a dysregulated SUMO system caused by loss of Ulp2. Loss of Ulp2 leads to accumulation of polySUMO-conjugated substrates, enhanced expression of genes encoding ribosomal proteins and reduced cell fitness (irregular cell outline). Duplication of ChrI and ChrXII provides a transient adaptive solution to amplify three protein-coding genes, *CCR4*, *CLN3*, and *CCW12*, and several snoRNA genes, *SNR61*, *SNR55*, and *SNR57*. Following evolution over many cell generations, disomies of both ChrI and XII are eliminated, concomitant with mutations in *UBC9*, *UBA2*, or *AOS1*. These SUMO-ligating enzyme mutations reduce SUMO conjugation and suppress the growth defects of *ulp2Δ* cells. Upregulation of numerous snoRNA genes appears to promote ribosome biogenesis, which is normally tightly regulated by the SUMO system

library plasmids and cloned into the pGP564 plasmid. Exceptions to this were the following: For pGP564-*REX3* + *SNR6* + *YLR108C*, pGP564-*YLR334C* + *tE(UUC)L* + *NUP2* or pGP564-*SGD1*, a 5-kb SacI fragment from 11-b5, 4.5-kb PstI fragment from 11-e11, and 5.5-kb *Sac*I fragment from 11-f11, respectively, were inserted into pGP564. pGP564-*REC102* was generated by religation of the 11-e11 vector fragment containing the *REC102* ORF after *Pst*I digestion. For the *ubc9* and *uba2* plasmids, the mutated ORF with ~0.8-kb of upstream and downstream sequences were amplified by PCR and cloned into pRS315. The amplified ORFs of *AOS1* or *aos1E346V* with ~0.8-kb of flanking sequences were cloned into pRS315-*UBA2*, pRS315-*uba2C162S* or YCplac33-*UBA2*. For overexpression of *HIS-SMT3*, the allele was amplified by PCR from MHY5339 and then cloned into p425-*GPD*. All constructs were confirmed by DNA sequencing.

**RNA isolation and RNA-seq analysis**. Total RNA was extracted as previously described[18]. Briefly, one $OD_{600}$ equivalent of cells grown to mid-exponential phase was used for RNA isolation using the RNeasy kit (Qiagen). The Ambion DNA-free DNA removal kit was used to remove DNA contaminants.

Twelve strand-specific sequencing libraries, three replicates per condition, were produced from the purified total RNA samples by the Illumina TruSeq stranded protocol. The libraries underwent 76-bp paired-end sequencing using an Illumina HiSeq 2500 according to Illumina protocols, generating an average of 14 million paired-end reads per library. For each read, we trimmed the first six nucleotides and the last nucleotides at the point where the Phred score of an examined base fell below 20 using in-house scripts. If, after trimming, the read was shorter than 45 bp, the entire read was discarded. Trimmed reads were mapped to the *S. cerevisiae* genome (Ensembl R64–1–1) with Tophat v2.1.1[57] using the very-sensitive preset, first-strand library type, and providing the corresponding gene model annotation. Only the reads that mapped to a single unique location within the genome, with a maximum of two mismatches in the anchor region of the spliced alignment, were reported in these results. We used the default settings for all other Tophat options. Tophat alignments were then processed by Cuffdiff (Cufflinks v2.2.1[58]) to obtain differential gene expression using first-strand library type, providing gene model annotation, and the genome sequence file for detection and correction of sequence-specific bias that random hexamer can cause during library preparation.

**Reverse transcription-polymerase chain reactions (RT-PCRs)**. One microgram of RNA was used in reverse transcription reactions using the iScript cDNA synthesis kit (Bio-Rad). The oligonucleotide sequences used for qPCR are listed in Supplementary Table 3. Diluted cDNA (1:200) was used in qPCR reactions to determine the expression of ribosomal protein genes *RPL12B* and *RPS18B*, snoRNA genes *SNR30* and *SNR60*, and the *ACT1* or *SPT15* controls. To measure expression of rRNA 18s and 25s, cDNA was diluted to 1:1,000. Each qPCR reaction was performed in technical triplicate. Relative RNA levels were calculated using the comparative Ct method ($\Delta\Delta$Ct)[59].

**DNA-sequencing (seq) analysis**. Genomic DNA was extracted from 1 $OD_{600}$ equivalent of cells using a standard phenol/chloroform extraction method.

Whole-genome sequencing was then performed by the Yale Center for Genome Analysis, using $2 \times 100$ bp sequencing on Illumina HiSeq 2500 instruments, to an average depth of 500x per sample. The reads were aligned to the *S. cerevisiae* genome using BWA MEM v0.7.15, and a joint calling of variants was performed using freebayes v0.9.14 with options (-P 0.5 -E 0 -q 10 -C 5).

**qPCR ploidy assay**. Chromosome copy number was determined by qPCR ploidy assays according to the protocol of Pavelka et al.[60] as described previously[18]. Briefly, DNA samples were prepared by the method described above. The oligonucleotide sequences used for qPCR are listed in Supplementary Table 3. Triplicate qPCR reactions were performed using the iQ SYBR Green Supermix kit in a Roche LightCycler 480 instrument, and the qPCR results were analyzed by the modified Ct method[59]. Ct values for each chromosome were normalized to the median Ct value for each cell, and then $\Delta$Ct was represented as relative ratio to WT control. For screening assays in Supplementary Fig. 6, each reaction was performed in technical duplicate.

**Chromatin immunoprecipitation (ChIP)**. ChIP experiments were done using established protocols[18,61]. Briefly, cells grown to mid-log phase in 200 ml of SD medium with suitable supplements were cross-linked with formaldehyde for 20 min and quenched with glycine. Cell pellets were lysed with glass beads and then sonicated to produce DNA fragments of ~500 bp. TAP-tagged, Flag-tagged and His-tagged proteins in clarified extracts were precipitated with 20 µl of IgG-Sepharose beads (17-0969-01, GE Healthcare), anti-Flag agarose beads (A2220, Sigma) and Ni-NTA agarose beads (R90115, Thermo Scientific), respectively. Following washings, eluted chromatin fragments were treated with pronase (Roche), and DNA was purified by phenol/chloroform extraction. The oligonucleotide sequences used for ChIP qPCR are listed in Supplementary Table 3. qPCR assays were performed using diluted template DNA (1:8 dilution for IP DNA and 1:1000 dilution for input DNA), and signals were normalized to the internal control (a fragment amplified from an untranscribed region on ChrIV) and input DNA.

**Immunoblotting**. Preparation of yeast whole-cell extracts and immunoblotting were carried out as described previously[62]. Levels of the Ulp2-AID∗-6Flag fusion protein, SUMO conjugate profiles, and Pgk1 were analyzed by immunoblotting with 1:2000 dilutions of anti-Flag (F3165, Sigma), anti-SUMO[63] and anti-Pgk1 (459250, Molecular Probes) antibodies, respectively. Sumoylated Rap1 was detected according to previous reports with little modification[36]. Specifically, TCA-extracted proteins from 50 $OD_{600}$ of cells were resuspended in 0.6 ml of sodium dodecyl sulphate (SDS)-sample buffer with 50 µl of unbuffered 2 M Tris, and then heated to 100 °C for 10 min. After centrifugation, 0.4 ml of supernatants were diluted with 0.8 ml of buffer (50 mM Tris HCl [pH 7.4], 5 mM EDTA, 150 mM NaCl, 1% Triton-X100) containing 20 mM NEM, immunoprecipitated with anti-Flag agarose beads (A2220, Sigma) for 4 h at 4 °C and then finally eluted into SDS-sample buffer by boiling for 5 min. Both IP and INPUT samples were analyzed by sodium dodecyl sulphate-polyacrylamide gel electrophoresis (SDS-PAGE) and

immunoblotting as described above. For analysis of ribosomal proteins, TCA-extracted proteins were separated by Tricine-SDS-PAGE and then detected by immunoblotting using 1:1000 of anti-Rpl32 antibody (PA5–69070, Invitrogen). Uncropped blots are presented in a Supplementary Fig. 11.

**Flow cytometry analysis**. Yeast DNA content was analyzed by flow cytometry as previously described[18,64]. Cells in exponential growth were fixed in 70% ethanol, digested with RNase A, and then stained with propidium iodide at 50 µg/ml final concentration. The analysis was carried out using a FACS LSRII flow cytometer (BD Biosciences) with a green laser (150 mW at 532 nm). Data were analyzed using FlowJo ver. 8 (Tree Star) and illustrated with two peaks indicating DNA contents, 1 C (unreplicated DNA) and 2 C (replicated DNA).

**Construction of ULP2-AID\*-6FLAG-neo-HHR-B2**. Directly after the stop codon in the gene for the Ulp2-AID\*-6FLAG fusion protein[65] is a 21-bp linker, the *neo-HHR-B2* sequence (91 bp), a 19-bp A-rich linker[56] and the authentic *ULP2* 3′UTR sequence. Marker-free genome editing was achieved by CRISPR/Cas9-mediated cleavage within the original *ULP2* stop codon (TAG) and insertion of a 545-bp insert with the aforementioned allele. Specifically, a PCR fragment including the 545 bp AID\*-6FLAG-neo-HHR-B2 cassette plus a 5′ flanking region from *ULP2* ORF nucleotides 3058–3102 and a 3′ flanking region from *ULP2* 3′UTR nucleotides 1–45 was amplified from plasmid pHyg-AIDstar-6FLAG_neo_HHR_B2 and used for integration. The pHyg-AIDstar-6FLAG_neo_HHR_B2 plasmid was generated by inserting the neo_HHR_B2 sequence amplified from plasmid pBT3-K2–3′fRzSu-AG-lib4-B2[65] (kind gift of J. Hartig, U. Konstanz) into XbaI/BglII-cut pHyg-AIDstar-6FLAG[56].

A one-vector system with sgRNA and Cas9 expression cassettes in a single plasmid[66] was used to direct genomic integration. A duplex of STK#58 (GATCT-ATGCGTGCGAGTGCTTTCAGTTTTAGAGCTAG) and STK#59 CTAGCTCTAAAACT-GAAAGCACTCGCACGCATA) oligonucleotides was cloned into SwaI,BclI-cut pML104;[66] the resulting pML104-gRNA-ULP2CT plasmid encodes the sgRNA that directs *ULP2* cleavage. The 635 bp PCR fragment and pML104-gRNA-ULP2CT were co-transformed into Y884 yeast. The desired insert was confirmed by PCR and DNA sequencing. The URA3-marked pML104-gRNA-ULP2CT plasmid was then evicted on a 5-FOA, resulting in yeast strain Y911. Y884 was created by integrating MscI-linearized pRS303-ADH1-AFB2 (kind gift of Helle Ulrich, IMB Mainz) at the *his3Δ1* locus of strain BY4741. The strain expresses the *A. thaliana* F-box protein AFB2 from the yeast *ADH1* promoter.

## Data availability

The RNA-seq and DNA-seq data used in this publication have been deposited in Gene Expression Omnibus (GEO) with accession GSE121898 and GSE121899, respectively. A reporting summary for this Article is available as a Supplementary Information file. All other data supporting the findings of this study are available from the corresponding authors on reasonable request.

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

## Acknowledgements

We thank Christopher Hickey for comments on the manuscript. This work was supported by NIH grant R01 GM053756 to M.H.

## Author contributions

H.Y.R. contributed to the experimental design, performed most of experiments and drafted the manuscript. F.L.G. and J.K. analyzed the RNA-seq and DNA-seq data, respectively. S.S.H. carried out flow cytometry analysis. C.R. and S.G.K. made the strain used for the Ulp2-AID* depletion analysis. M.H. designed the study, analyzed the data, and wrote the final version of the manuscript.
