## [Peer Review File · Nature Communications]

Reviewers' comments:

Reviewer #1 (Remarks to the Author):

Yona et al. proposed several years ago that aneuploidy provides a rapid, yet transient, solution to acute growth defects in *S. cerevisiae*. Data consistent with this model has appeared since, but few studies have addressed the question directly. Hochstrasser and colleagues previously found that *S. cerevisiae* mutants that, following loss of Ulp2 are often rescued by aneuploidies of chromosomes I and XII, and they found two genes on Chromosome I, CLN3 and CCR4, can explain the rescue phenotype. In this paper they show that three snoRNA genes on Chr12 are likely to contribute to the rescue phenotype and show that if exogenous copies of these three snoRNAs are provided on a plasmid, the *ulp2* mutant no longer becomes aneuploid for Chr12. Importantly, after further selection for growth, they now report that the aneuploidies can be lost, sequentially, and new point mutations enabling survival are acquired in the absence of the rescuing aneuploidies. In particular they found the sumo ligating enzyme Ubc9. In an additional, directed search for other enzymes in the sumo pathway, they found point mutations in Uba2. These two suppressor mutations, when present in the starting strain that loses ULP2, are sufficient to prevent the appearance of ChrXII aneuploidy. Furthermore, they provide clear explanations for how the original mutation is suppressed in that they directly modify the sumolation process that was originally defective when Ulp2 was lost. The mechanisms by which other suppressors, such as cell cycle checkpoint genes, BUB1 and BUB3, MAD3, as well as other genes (e.g., IFM1, SPO23) is not discussed in any detail.

In general, the paper is well written, the work is of general interest and the results help solidify the idea of transient aneuploidy preceding precise, highly effective mutational solutions. There are a few issues that should be addressed, most of them minor.

1. Introduction lines 81-108 are a bit jumbled—the order of the ideas is not as clear as it could be and the subject jumps from yeast to mammalian systems and back again abruptly. This can be solved with some attention to carefully laying out the concepts in a logical order.
2. The figures were very small and it was often difficult to read the axis labels (e.g., Fig. 1C).
3. The authors should consider framing the work in the evolutionary context laid out by Yona et al. 2012.
4. Given that chr1 disomy increases the copy number of Cln3 (a G1/S cyclin), could the cell cycle checkpoint genes identified here be providing a solution to the lack of Ulp2 that is more akin to the solution provided by Chr1 disomy?

Reviewer #2 (Remarks to the Author):

The authors show that loss of the SUMO protease Ulp2p leads to chromosome I and XII disomy. Prolonged culturing of *ulp2* delta cells leads to restoration of normal chromosome I and XII copy number after approximately 100 and 500 generations, respectively. Nascent *ulp2* null cells display increased expression of many ribosomal protein (RP) genes, as well as growth and cell cycle defects. In long term adapted *ulp2* delta cells, quasi normal growth, cell cycle and RP gene expression are restored. In contrast, expression of more than half of cellular snoRNAs is elevated.

The search for genes on chromosome XII whose overexpression could suppress chromosome XII disomy of nascent *ulp2* delta cells led to the surprising discovery that extra copies of a chromosome XII fragment harbouring three box C/D snoRNAs (snR61, snR55 and snR57) could prevent such disomy. Several long term adapted *ulp2* delta strains have acquired mutations in UBC9, encoding a SUMO-conjugating E2 enzyme, or in UBA2, encoding the catalytic subunit of the heterodimeric SUMO-activating enzyme E1, or in AOS1, encoding the other E1 enzyme subunit. These long term adapted *ulp2* delta strains with mutations in SUMO pathway components display strongly decreased polySUMO conjugate levels compared to nascent *ulp2* delta cells. A few long term adapted *ulp2* delta strains seem devoid of mutations in SUMO pathway components, consistent with the fact that these strains maintain high levels of polySUMO conjugates. The present article contains an impressive amount of data and highlights the complex mechanisms involved in short and long term adaptation to severe perturbation in SUMO-protein conjugation dynamics. The work will be of high interest to scientists dealing with post-transcriptional modifications, stress adaptation and ribosome biogenesis.

Specific points:

- The authors should check the levels of the mature forms of a subset of snoRNAs by Northern blot experiments in wild type and *ulp2* delta cells grown for 50 and 500 generations.
- The authors checked ten separate *ulp2* delta lines for chromosome I and XII disomy, cell cycle defects and accumulation of polySUMO conjugates (pages 13-14). As far as I could tell, they did not report effects on RP and snoRNA transcript levels. Did these additional *ulp2* delta lines also display transient increase in RP transcript levels followed by an increase in snoRNA transcript levels, as reported for the first *ulp2* delta line described in this paper ?
- The authors should check that the overexpression of the mature forms of the snR61, snR55 and snR57 snoRNAs is actually responsible for the suppression of the chromosome XII disomy: they should analyse the levels of the mature snoRNAs by Northern in *ulp2* delta cells transformed with the plasmid containing the snoRNA cluster and should check that point mutations in conserved boxes of the snoRNAs (that should abolish the accumulation of the mature forms of the snoRNAs) prevent the suppression of the chromosome XII disomy.
- How do the authors reconcile the fact that chromosome XII disomy (with thus increased dosage of snR61, snR55 and snR57 snoRNA genes) in nascent *ulp2* delta cells is accompanied by an increase of RPL12B and RPS18B transcript levels while the plasmid-encoded snR61, snR55 and snR57 snoRNA gene cluster suppresses the increased expression of these RP genes. Is it a question of the actual level of snoRNA overexpression ? Please clarify and/or discuss.
- Did the authors identify the mutations in the three long term adapted *ulp2* delta strains devoid of mutations in SUMO pathway components, that maintain high levels of polySUMO conjugates ?
- Are the snoRNA genes that are targeted by Ulp2p precisely those that are upregulated in long term adapted *ulp2* delta strains ?

Reviewer #3 (Remarks to the Author):

This study from the Hochstrasser laboratory examined cellular evolution upon Ulp2 loss. They defined two stages of cellular adaptation under this condition – a transient aneuploidy stage with specific gain of chr I and XII; this then gives way to a long-term state wherein compensatory mutations reducing sumoylation level in combination of specific snoRNA increase can restore cell homeostasis in the absence of Ulp2. RNA-seq data showed wide-spectrum of changes upon Ulp2 loss. Among these, RP genes as a group showed temporal increase at 50 generation (G) and back

to normal at 500G. This may be due to a previous reported effect of Rap1 sumoylation in promoting RP gene transcription. On the other hand, snoRNAs as a group showed increased levels at 500G only. A few tested snoRNA can reduce RP RNA levels, suggesting inverse relationship. CHIP analyses show that Ulp2 is present at RP and snoRNA promoters, while Hst3/Nrg2 whose RNA levels were downregulated in *ulp2Δ* cells did not show an overall association at these promoters. As predicted, *ulp2* loss increase SUMO CHIP signals at these promoters. The authors conclude that Ulp2 can directly suppress RP/snoRNA transcription. It was unclear whether this function is relevant for successful long-term adaptation or just one of the multiple immediate consequences of *ulp2* loss. In a parallel line of study, it was found that extra copies of CCW2, a mannoprotein involved in cell wall function, and three snoRNA, can bypass Chr XII aneuploidy in *ulp2* loss situation. The latter suppresses the excess RP gene expression in nascent *ulp2* mutant cells. When examining *ulp2Δ* cells after 500G passage, aneuploidy disappeared, and extra mutations were acquired. Among these mutations were those affecting SUMO E2 (K27N) and E1 (several mutants) that can reduce sumoylation without affecting cell growth and can prevent disomy formation at the low pass *ulp2Δ* population.

This is an interesting study that elucidates how cells adapt to increased sumoylation levels through two distinct stages, a short-term stage with chr I and XII disomy, and a long-term stage with SUMO E1/E2 mutation to reset the sumoylation homeostasis. Along this journey, RP and snoRNA expression level changes in the opposite direction may play some roles, though further understanding of this possibility can enhance the impact of this work. I have a few suggestions to improve clarity and mechanistic understanding.

- 1) Is it known that Ulp2 is responsible for removing sumoylation from Rap1? Along the same line, does Rap1 sumoylation status affect the adaption process; for example, perhaps Rap1 sumoylation mutants impair the adaption process in *ulp2Δ* cells or the ability to generate SUMO E2 mutants?
- 2) Like RP genes, snoRNA genes also were enriched with Ulp2 CHIP signals, however, the transcriptional levels of these two group of genes behave differently. How to explain this difference?
- 3) The role of snoRNA55/57/61 in the adaptation to *ulp2* loss is interesting but not completely clear. Does *ulp2Δ* affect snoRNA55/57/61 expression and do Ulp2 CHIP signals present at their promoters, given that *ulp2* loss increased the expression of some snoRNAs and Ulp2 was present at some snoRNA promoters? Figure 4g showed that extra copies of snoRNA55/57/61 led to 2-fold reduction of RPL12B and RPS18B transcription, does this change bear biological significance considering that several other snoRNA can have similar effects (figure 2g). Also, it is unclear if the change of RPL12B and RPS18B transcription really lead to protein level increase.
- 4) Does increasing snoRNA55/57/61 expression, which bypasses the short-term adaptation stage of Chr XII disomy, mitigates the need or ability of long-term adaptation in *ulp2Δ* cells?

Point-by-point response to reviewers' comments:

Reviewer #1 (Remarks to the Author):

Yona et al. proposed several years ago that aneuploidy provides a rapid, yet transient, solution to acute growth defects in *S. cerevisiae*. Data consistent with this model has appeared since, but few studies have addressed the question directly. Hochstrasser and colleagues previously found that *S. cerevisiae* mutants that, following loss of Ulp2 are often rescued by aneuploidies of chromosomes I and XII, and they found two genes on Chromosome I, CLN3 and CCR4, can explain the rescue phenotype. In this paper they show that three snoRNA genes on Chr12 are likely to contribute to the rescue phenotype and show that if exogenous copies of these three snoRNAs are provided on a plasmid, the *ulp2* mutant no longer becomes aneuploid for Chr12. Importantly, after further selection for growth, they now report that the aneuploidies can be lost, sequentially, and new point mutations enabling survival are acquired in the absence of the rescuing aneuploidies. In particular they found the sumo ligating enzyme Ubc9. In an additional, directed search for other enzymes in the sumo pathway, they found point mutations in Uba2. These two suppressor mutations, when present in the starting strain that loses ULP2, are sufficient to prevent the appearance of ChrXII aneuploidy. Furthermore, they provide clear explanations for how the original mutation is suppressed in that they directly modify the sumoylation process that was originally defective when Ulp2 was lost. The mechanisms by which other suppressors, such as cell cycle checkpoint genes, BUB1 and BUB3, MAD3, as well as other genes (e.g., IFM1, SPO23) is not discussed in any detail.

In general, the paper is well written, the work is of general interest and the results help solidify the idea of transient aneuploidy preceding precise, highly effective mutational solutions. There are a few issues that should be addressed, most of them minor.

We thank the reviewer for the positive review and constructive comments.

1. Introduction lines 81-108 are a bit jumbled—the order of the ideas is not as clear as it could be and the subject jumps from yeast to mammalian systems and back again abruptly. This can be solved with some attention to carefully laying out the concepts in a logical order.

To make this clearer, we have removed the somewhat extraneous discussion of mammalian data here and moved the description of Rap1 sumoylation to the Results now that we have direct data on Rap1-SUMO.

2. The figures were very small and it was often difficult to read the axis labels (e.g., Fig. 1C).

In order to improve this problem, we've increased the font size from 11 pt to 13 pt in all figures and re-organized panels in some places to make viewing easier.

3. The authors should consider framing the work in the evolutionary context laid out by Yona et al. 2012.

Yona et al. followed a more complex pipeline for their laboratory evolution experiments, making it difficult for us to exactly follow their framework. However, we have added the phrase “laboratory evolution” to the diagram in Fig. 1a to follow their diagram.

4. Given that chr1 disomy increases the copy number of Cln3 (a G1/S cyclin), could the cell cycle checkpoint genes identified here be providing a solution to the lack of Ulp2 that is more akin to the solution provided by Chr1 disomy?

This is an interesting possibility. We note that the cell cycle checkpoint gene mutations in *BUB1*, *MAD3* and *BUB3* were *only* identified in *ulp2Δ* cells still containing a WT *ULP2* plasmid during passaging (*ulp2Δ* +p*ULP2* 50G or 500G). These are quasi-WT cells, but fluctuations in Ulp2 levels could lead to cell cycle arrest in response to checkpoint activation (a known effect of losing Ulp2); this could create selective pressure for mutations in checkpoint genes. We have rewritten this section to clarify the link to possible checkpoint arrest and cited Schwartz et al. (2007) in support of this.

Reviewer #2 (Remarks to the Author):

The authors show that loss of the SUMO protease Ulp2p leads to chromosome I and XII disomy. Prolonged culturing of *ulp2* delta cells leads to restoration of normal chromosome I and XII copy number after approximately 100 and 500 generations, respectively. Nascent *ulp2* null cells display increased expression of many ribosomal protein (RP) genes, as well as growth and cell cycle defects. In long term adapted *ulp2* delta cells, quasi normal growth, cell cycle and RP gene expression are restored. In contrast, expression of more than half of cellular snoRNAs is elevated. The search for genes on chromosome XII whose overexpression could suppress chromosome XII disomy of nascent *ulp2* delta cells led to the surprising discovery that extra copies of a chromosome XII fragment harbouring three box C/D snoRNAs (snR61, snR55 and snR57) could prevent such disomy. Several long term adapted *ulp2* delta strains have acquired mutations in UBC9, encoding a SUMO-conjugating E2 enzyme, or in UBA2, encoding the catalytic subunit of the heterodimeric SUMO-activating enzyme E1, or in AOS1, encoding the other E1 enzyme subunit. These long term adapted *ulp2* delta strains with mutations in SUMO pathway components display strongly decreased polySUMO conjugate levels compared to nascent *ulp2* delta cells. A few long term adapted *ulp2* delta strains seem devoid of mutations in SUMO pathway components, consistent with the fact that these strains maintain high levels of polySUMO conjugates. The present article contains an impressive amount of data and highlights the complex mechanisms involved in short and long term adaptation to severe perturbation in SUMO-protein conjugation dynamics. The work will be of high interest to scientists dealing with post-transcriptional modifications, stress adaptation and ribosome biogenesis.

We thank the referee for the positive overall evaluation and the constructive comments to extend several points in the paper.

Specific points:

- The authors should check the levels of the mature forms of a subset of snoRNAs by Northern blot experiments in wild type and *ulp2* delta cells grown for 50 and 500 generations.

This is a good suggestion. However, we recently obtained interesting results that can help explain the shifts in RP and snoRNA levels at early and late passaging based on effects of Ulp2 and Ccr4 on snoRNA and RP gene transcription. Therefore, we have focused on those novel findings and added them to Fig. 3.

Briefly, we found that levels of Ccr4 protein, a subunit of the Ccr4-Not complex, were strongly enhanced due to the *CCR4* gene duplication caused by ChrI disomy in *ulp2Δ* cells (new Fig. 3f). This increase is accompanied by enhanced association of Ccr4 with the promoters of snoRNA (and rDNA), but not of RP genes (Fig. 3g). This binding of Ccr4 is linked to the suppressed expression of snoRNAs and rRNA in *ulp2Δ* cells (Fig. 3d, e). Therefore, we suggest that both Ccr4 and Ulp2 may generally inhibit transcription of snoRNAs. Following many cell generations and the loss of the extra ChrI in *ulp2Δ* cells, cellular levels of Ccr4 renormalize, allowing snoRNA expression to increase (Fig. 2).

- The authors checked ten separate *ulp2* delta lines for chromosome I and XII disomy, cell cycle defects and accumulation of polySUMO conjugates (pages 13-14). As far as I could tell, they did not report effects on RP and snoRNA transcript levels. Did these additional *ulp2* delta lines also display transient increase in RP transcript levels followed by an increase in snoRNA transcript levels, as reported for the first *ulp2* delta line described in this paper ?

These are good points. In response, we have now checked the levels of RP and snoRNA transcripts in the ten independent *ulp2Δ* lines at 500 generations by qRT-PCR (Fig. 5d). Consistent with RNA-seq data obtained from the original *ulp2Δ* strain at 500 generations in Fig. 2, all of these *ulp2Δ* strains showed a reduction of transcripts for the RP genes *RPL12B* and *RPS18B* back to normal levels, and two-fold or greater increase in levels of the snoRNAs *SNR30* and *SNR60* relative to WT.

- The authors should check that the overexpression of the mature forms of the snR61, snR55 and snR57 snoRNAs is actually responsible for the suppression of the chromosome XII disomy: they should analyse the levels of the mature snoRNAs by Northern in *ulp2* delta cells transformed with the plasmid containing the snoRNA cluster and should check that point mutations in conserved boxes of the snoRNAs (that should abolish the accumulation of the mature forms of the snoRNAs) prevent the suppression of the chromosome XII disomy.

We agree that our data do not formally show that overexpression of the mature forms of snR61, snR55, and snR57 snoRNAs are capable of suppressing the chrXII disomy. The experiments suggested are good ones, but we feel they go beyond the current study. Our original data did show clearly that increased gene dosage of the snR61-snR55-snR57 snoRNA gene cluster prevents the ChrXII disomy and substantially suppresses the high levels of RP mRNAs normally found in nascent *ulp2Δ* cells. We have now added new ChIP data showing association of Ulp2-Flag with *SNR57* and enhanced levels of SUMO at the *SNR57* locus when Ulp2 is lost (Fig. 4f, g), consistent with a transcriptional role for Ulp2 at snoRNA gene loci.

- How do the authors reconcile the fact that chromosome XII disomy (with thus increased dosage of snR61, snR55 and snR57 snoRNA genes) in nascent *ulp2* delta cells is accompanied by an increase of RPL12B and RPS18B transcript levels while the plasmid-encoded snR61, snR55 and snR57 snoRNA gene cluster suppresses the increased expression of these RP genes. Is it a question of the actual level of snoRNA overexpression? Please clarify and/or discuss.

We have added the phrase “the much higher dosage of these three snRNA genes on the plasmid compared to ChrXII disomy presumably accounts for the suppression of RP transcripts only in the former case” near the bottom of page 12 to clarify this.

- Did the authors identify the mutations in the three long term adapted *ulp2* delta strains devoid of mutations in SUMO pathway components, that maintain high levels of polySUMO conjugates?

We are also extremely interested in determining the basis for long-term adaptation in these three strains. We have begun DNA-seq and RNA-seq analyses as a first step to try to identify the molecular changes in them. At present, we do not have a clear understanding of the relevant adaptive mutations in any of the three cases. This work will be part of our future studies.

- Are the snoRNA genes that are targeted by Ulp2p precisely those that are upregulated in long term adapted *ulp2* delta strains?

Although we have not analyzed all 77 snoRNA genes by ChIP and qRT-PCR assays, we do observe that Ulp2 is directly targeted at least to all three tested snoRNA genes that were upregulated in *ulp2Δ* 500G cells (Fig. 2), *SNR30*, *SNR40* and *SNR60*. Furthermore, we now also show data that the transcript levels of *SNR30* and *SNR60* were significantly increased in a *ulp2Δ ccr4Δ* double mutant, compared with *ulp2Δ* (Fig. 3e), suggesting these two snoRNA genes, at least, are targets for repression by both Ulp2 and Ccr4. Therefore, we infer that snoRNAs are likely to be general targets of Ulp2.

Reviewer #3 (Remarks to the Author):

This study from the Hochstrasser laboratory examined cellular evolution upon Ulp2 loss. They defined two stages of cellular adaptation under this condition – a transient aneuploidy stage with specific gain of chr I and XII; this then gives way to a long-term state wherein compensatory mutations reducing sumoylation level in combination of specific

snoRNA increase can restore cell homeostasis in the absence of Ulp2. RNA-seq data showed wide-spectrum of changes upon Ulp2 loss. Among these, RP genes as a group showed temporal increase at 50 generation (G) and back to normal at 500G. This may be due to a previously reported effect of Rap1 sumoylation in promoting RP gene transcription. On the other hand, snoRNAs as a group showed increased levels at 500G only. A few tested snoRNA can reduce RP RNA levels, suggesting inverse relationship. ChIP analyses show that Ulp2 is present at RP and snoRNA promoters, while Hst3/Nrg2 whose RNA levels were downregulated in *ulp2Δ* cells did not show an overall association at these promoters. As predicted, *ulp2* loss increase SUMO ChIP signals at these promoters. The authors conclude that Ulp2 can directly suppress RP/snoRNA transcription. It was unclear whether this function is relevant for successful long-term adaptation or just one of the multiple immediate consequences of *ulp2* loss. In a parallel line of study, it was found that extra copies of CCW2, a mannoprotein involved in cell wall function, and three snoRNA, can bypass Chr XII aneuploidy in *ulp2* loss situation. The latter suppresses the excess RP gene expression in nascent *ulp2* mutant cells. When examining *ulp2Δ* cells after 500G passage, aneuploidy disappeared, and extra mutations were acquired. Among these mutations were those affecting SUMO E2 (K27N) and E1 (several mutants) that can reduce sumoylation without affecting cell growth and can prevent disomy formation at the low pass *ulp2Δ* population.

This is an interesting study that elucidates how cells adapt to increased sumoylation levels through two distinct stages, a short-term stage with chr I and XII disomy, and a long-term stage with SUMO E1/E2 mutation to reset the sumoylation homeostasis. Along this journey, RP and snoRNA expression level changes in the opposite direction may play some roles, though further understanding of this possibility can enhance the impact of this work. I have a few suggestions to improve clarity and mechanistic understanding.

We thank the reviewer for the positive review and constructive comments.

1) Is it known that Ulp2 is responsible for removing sumoylation from Rap1? Along the same line, does Rap1 sumoylation status affect the adaptation process; for example, perhaps Rap1 sumoylation mutants impair the adaptation process in *ulp2Δ* cells or the ability to generate SUMO E2 mutants?

This is a good suggestion. Therefore, we have now tested whether sumoylated Rap1 is a substrate of Ulp2 (Supplementary Fig. 5a). Indeed, our results show that levels of poly-SUMO conjugated Rap1 were clearly elevated in cells lacking *ULP2*. Association of Rap1 with RP and snoRNA genes, however, was not significantly affected by loss of Ulp2 (Supplementary Fig. 5b), consistent with data suggesting sumoylation of Rap1 functions downstream of Rap1 binding to RP promoters (*Genome Research*, 2015). The elevated levels of SUMO-conjugated Rap1 in *ulp2Δ* may promote the RP gene transcription and rapamycin resistance that we observed, but we have not yet tested Rap1 sumoylation mutants as this would extend beyond the goals of the present study.

2) Like RP genes, snoRNA genes also were enriched with Ulp2 ChIP signals, however, the

transcriptional levels of these two group of genes behave differently. How to explain this difference?

This is a good question and as outlined in our response to Reviewer #1, we believe our new data on Ccr4 can account for this difference (Fig. 3d, e, f, g). The new findings are summarized in the model in Figure 3h. Our results and previous findings suggest that Ulp2 may generally inhibit transcription of the RP, snoRNA and rRNA genes, whereas Ccr4 blocks the expression of snoRNA and rRNA genes, but not RP gene expression (Fig. 3h, left). The loss of Ulp2 leads to increased Ccr4 protein as a result of ChrI disomy, which thereby limits the transcription of snoRNA and rRNA genes (Fig. 3h, right). The net effect is a specific increase in RP expression in incipient *ulp2Δ* cells. These data could explain the differential changes of gene expression of RPs and snoRNAs in *ulp2Δ* cells.

3) The role of snoRNA55/57/61 in the adaptation to *ulp2* loss is interesting but not completely clear. Does *ulp2Δ* affect snoRNA55/57/61 expression and do Ulp2 ChIP signals present at their promoters, given that *ulp2* loss increased the expression of some snoRNAs and Ulp2 was present at some snoRNA promoters? Figure 4g showed that extra copies of snoRNA55/57/61 led to 2-fold reduction of RPL12B and RPS18B transcription, does this change bear biological significance considering that several other snoRNA can have similar effects (figure 2g). Also, it is unclear if the change of RPL12B and RPS18B transcription really lead to protein level increase.

These are good questions, and in the revised manuscript we have provided new data to address most of them:

First, we tested whether Ulp2 protein is localized at the chrXII snoRNA polycistronic SNR57/55/61 locus. As predicted from our earlier data (Fig. 3a), Ulp2 was directly recruited to both promoter and coding sequences of *SNR57* (Fig. 4f), and SUMO levels at these sites were increased in *ulp2Δ* cells (Fig. 4g), suggesting that Ulp2 is involved in the transcription of *SNR57* as well as other snoRNA genes.

Second, we have determined whether the elevated RP mRNA levels actually lead to increased protein expression in response to different high-copy snRNA plasmids. Because we do not have antibodies against Rpl12B or Rps18B, we performed immunoblotting using anti-Rpl32 antibody (new Fig. 2h). 11 of the original 13 plasmids as well as an additional plasmid with the three snoRNA genes from ChrXII reduced excess accumulation of Rpl32 protein upon loss of Ulp2. Therefore, we suggest that high expression of various different snoRNA genes normally relieves cellular defects accompanying high expression of ribosomal proteins in cells lacking *ULP2*.

4) Does increasing snoRNA55/57/61 expression, which bypasses the short-term adaptation stage of Chr XII disomy, mitigates the need or ability of long-term adaptation in *ulp2Δ* cells?

We think this is unlikely. In nascent *ulp2Δ* cells with a high-copy snoRNA55/57/61 plasmid, many cellular defects remain. ChrI disomy, growth defects and the excess accumulation of high molecular weight SUMO conjugates are not suppressed by increasing these snoRNAs (Fig. 4d

bottom and Supplementary Fig. 8). Therefore, we believe that these cells still would require additional long-term adaptive changes to achieve the near-WT phenotype seen in the high-passage strains.

REVIEWERS' COMMENTS:

Reviewer #2 (Remarks to the Author):

The authors have in my view answered in a satisfactory manner most of the issues raised by the reviewers and added a substantial amount of novel data to support their claims.

I am therefore in favour of accepting the revised manuscript.